# Matrix Product Sketching via Coordinated Sampling

**Majid Daliri[†], Juliana Freire[†], Danrong Li[∗], Christopher Musco[†]**
[†]New York University
[∗]Pennsylvania State University

## Abstract

We revisit the well-studied problem of approximating a matrix product, $\mathbf{A}^T\mathbf{B}$, based on small space sketches $\mathcal{S}(\mathbf{A})$ and $\mathcal{S}(\mathbf{B})$ of $\mathbf{A} \in \mathbb{R}^{n \times d}$ and $\mathbf{B} \in \mathbb{R}^{n \times m}$. We are interested in the setting where the sketches must be computed independently of each other, except for the use of a shared random seed. We prove that, when $\mathbf{A}$ and $\mathbf{B}$ are sparse, methods based on *coordinated random sampling* can outperform classical linear sketching approaches, like Johnson-Lindenstrauss Projection or CountSketch. For example, to obtain Frobenius norm error $\epsilon\|\mathbf{A}\|_F\|\mathbf{B}\|_F$, coordinated sampling requires sketches of size $O(s/\epsilon^2)$ when $\mathbf{A}$ and $\mathbf{B}$ have at most $s \leq d, m$ non-zeros per row. In contrast, linear sketching leads to sketches of size $O(d/\epsilon^2)$ and $O(m/\epsilon^2)$ for $\mathbf{A}$ and $\mathbf{B}$. We empirically evaluate our approach on two applications: 1) distributed linear regression in databases, a problem motivated by tasks like dataset discovery and augmentation, and 2) approximating attention matrices in transformer-based language models. In both cases, our sampling algorithms yield an order of magnitude improvement over linear sketching.

## 1 Introduction

Over the past 20 years, sketching and sampling methods have emerged as powerful tools for solving massive linear algebraic problems that arise in machine learning, data science, and scientific computing (Woodruff, 2014; Drineas & Mahoney, 2016; Martinsson & Tropp, 2020). Matrix and vector sketching has also been widely applied in federated learning (Rothchild et al., 2020; Konečný et al., 2020), distributed learning (Jiang et al., 2018), and beyond (Cohen et al., 2015a). One of the most fundamental problems where randomization has found success is matrix-matrix multiplication: we are given $\mathbf{A} \in \mathbb{R}^{n \times d}$ and $\mathbf{B} \in \mathbb{R}^{n \times m}$ and hope to compute an approximation to the product $\mathbf{A}^T\mathbf{B} \in \mathbb{R}^{d \times m}$. Naively, it takes $O(dnm)$ time to compute $\mathbf{A}^T\mathbf{B}$ exactly, or a bit less if fast (rectangular) matrix multiplication methods are used (Le Gall, 2012). We are also interested in minimizing communication costs: when $\mathbf{A}$ and $\mathbf{B}$ are stored in separate machines, computing $\mathbf{A}^T\mathbf{B}$ requires communicating at least $O(d \cdot \min(n, m))$ numbers (either all of $\mathbf{A}$ or all of $\mathbf{B}$). We explore the question of whether this bound can be improved.

In particular, we consider solving matrix-matrix multiplication in the widely studied *sketching setting* (Nelson & Nguyen, 2013). The goal is to compute small space sketches $\mathcal{S}(\mathbf{A})$ and $\mathcal{S}(\mathbf{B})$, from which the matrix product can be approximated using some routine $\mathcal{F}(\mathcal{S}(\mathbf{A}), \mathcal{S}(\mathbf{B})) \approx \mathbf{A}^T\mathbf{B}$. Ideally, $\mathcal{S}(\mathbf{A})$ and $\mathcal{S}(\mathbf{B})$ should be much smaller than the space required to store $\mathbf{A}$ and $\mathbf{B}$ – i.e. smaller than $O(dn)$ and $O(nm)$ space for dense matrices, or smaller than $O(\text{nnz}(\mathbf{A}))$ and $O(\text{nnz}(\mathbf{B}))$ space when the matrices are sparse with nnz denoting the number of non-zeros. This reduction in size allows the sketches to be processed, stored, and communicated with less cost than the original matrices.

We emphasize that, in the sketching setting, while $\mathcal{S}(\mathbf{A})$ and $\mathcal{S}(\mathbf{B})$ can be computed using a shared source of random bits (e.g., for constructing hash functions), no additional communication is allowed between the processes computing $\mathcal{S}(\mathbf{A})$ and $\mathcal{S}(\mathbf{B})$. This restriction is crucial in settings where communication between processes is expensive, slow, or impossible, as is often the case in distributed systems, or in applications where $\mathbf{A}$ and $\mathbf{B}$ are processed at different times. Moreover, the restriction is critical in applications that require repeated matrix multiplications. For example, if we want to approximate $\mathbf{A}^{1T}\mathbf{B}, \ldots, \mathbf{A}^{qT}\mathbf{B}$ for a set of matrices $\mathbf{A}^1, \ldots, \mathbf{A}^q$ using the same sketch $\mathcal{S}(\mathbf{B})$, that

sketch cannot be tailored to any one particular $\mathbf{A}^i$. We further discuss applications of matrix-product sketching to central problems like dataset discovery and multi-vector retrieval in Section 1.4.

## 1.1 Prior Work

The best existing sketching methods for matrix product approximation are based on *linear sketches* that compress $\mathbf{A}$ and $\mathbf{B}$ via multiplication by a random matrix generated with a shared random seed. In particular, below we state a seminal result of Sarlós that was later refined and generalized (see Cohen et al. (2016) or Woodruff (2014) for more detailed discussions).

**Fact 1** ((Sarlós, 2006; Kane & Nelson, 2014; Cohen et al., 2016)). *Let $\mathbf{\Pi} \in \mathbb{R}^{k \times n}$ be a scaled random Gaussian matrix, random sign matrix, CountSketch matrix (Charikar et al., 2002), or any of a variety of other randomized linear embeddings. If $k = O\left(\frac{1}{\epsilon^2 \delta}\right)$, then with probability at least $1 - \delta$,*

$$\|(\mathbf{\Pi}\mathbf{A})^T(\mathbf{\Pi}\mathbf{B}) - \mathbf{A}^T\mathbf{B}\|_F \leq \epsilon\|\mathbf{A}\|_F\|\mathbf{B}\|_F.$$

$\mathbf{\Pi}\mathbf{A} \in \mathbb{R}^{k \times d}$ *and* $\mathbf{\Pi}\mathbf{B} \in \mathbb{R}^{k \times m}$ *are sketches of size* $O(d/\epsilon^2\delta)$ *and* $O(m/\epsilon^2\delta)$ *respectively. For dense matrices, these sketches are smaller than* $\mathbf{A}$ *and* $\mathbf{B}$*, respectively, whenever* $\frac{1}{\epsilon^2\delta} < n$.

The above result can be strengthened if additional assumptions are made on $\mathbf{A}$ and $\mathbf{B}$. For example, a tighter bound is possible if the matrices are low-rank or nearly low rank (Cohen et al., 2016). However, if no additional assumptions are made on the matrices' spectra, then Fact 1 is the best result known.

## 1.2 Sampling Based Matrix Product Approximation

Our goal is to devise an alternative approach to matrix product sketching that improves on Fact 1 in the important setting where $\mathbf{A}$ and $\mathbf{B}$ are *sparse*, providing better sketches even for dense matrices. To do so, we build on another classical approach for using randomization to speed up matrix-matrix multiplication: subsampling. A seminal paper by Drineas, Kannan, and Mahoney (Drineas & Kannan, 2001; Drineas et al., 2006a) proves that it is possible to sample and reweight $O(1/\epsilon^2\delta)$ rows of $\mathbf{A}$ and $\mathbf{B}$ so that the product of the subsampled matrices $\tilde{\mathbf{A}}$ and $\tilde{\mathbf{B}}$ satisfies $\|\tilde{\mathbf{A}}^T\tilde{\mathbf{B}} - \mathbf{A}^T\mathbf{B}\|_F \leq \epsilon\|\mathbf{A}\|_F\|\mathbf{B}\|_F$, with probability at least $1 - \delta$.

Notably, this approximation guarantee exactly matches Fact 1. However, since $\tilde{\mathbf{A}}$ and $\tilde{\mathbf{B}}$ consist of a *subsample* of rows from $\mathbf{A}$ and $\mathbf{B}$, they can be much more compact to store than $\mathbf{\Pi}\mathbf{A}$ and $\mathbf{\Pi}\mathbf{B}$. For example, if $\mathbf{A}$ and $\mathbf{B}$ have at most $s$ non-zeros per row, $\tilde{\mathbf{A}}$ and $\tilde{\mathbf{B}}$ take $O(s/\epsilon^2)$ space to store, instead of $O(d/\epsilon^2)$ and $O(m/\epsilon^2)$, respectively.

Importantly, however, the row subsamples $\tilde{\mathbf{A}}$ and $\tilde{\mathbf{B}}$ guaranteed by Drineas et al. (2006a) *cannot be computed in the sketching setting*. The challenge is that, to obtain a theoretical accuracy bound, rows must be sampled with non-uniform probabilities. Intuitively, if $\mathbf{A}$'s $i^{\text{th}}$ row $\mathbf{A}_i \in \mathbb{R}^d$ and $\mathbf{B}$'s $i^{\text{th}}$ row $\mathbf{B}_i \in \mathbb{R}^m$ have large magnitude, they contribute more to the matrix product $\mathbf{A}^T\mathbf{B}$, which can be written as a sum of rank-one outerproducts: $\mathbf{A}^T\mathbf{B} = \sum_{i=1}^n \mathbf{A}_i\mathbf{B}_i^T$. So, high-magnitude rows must be sampled with higher probability. The original analysis from Drineas et al. (2006a) suggests sampling with probabilities proportional to $\sim \|\mathbf{A}_i\|_2\|\mathbf{B}_i\|_2$, where $\|\mathbf{x}\|_2$ denotes the Euclidean norm of a vector $\mathbf{x}$. While these probabilities were shown to satisfy an optimal variance property, they cannot be computed without access to *both* $\mathbf{A}$ and $\mathbf{B}$, which is impossible in the sketching setting, since the sketches $\mathcal{S}(\mathbf{A})$ and $\mathcal{S}(\mathbf{B})$ must be computed independently of each other. It can be shown that it also suffices to use probabilities proportional to either $\sim \|\mathbf{A}_i\|_2^2$ or $\sim \|\mathbf{B}_i\|_2^2$ (so, only taking one matrix into account), but the choice must be consistent, so the issue remains that either the machine sketching $\mathbf{A}$ or the machine sketching $\mathbf{B}$ does not know the right probabilities.

We emphasize that this issue cannot simply be resolved by communicating probabilities between the processes computing the matrix sketches (which would be relatively inexpensive). In particular, the applications we consider in Section 1.4 require computing all pairwise matrix products between two sets of matrices $\mathbf{A}^1, \ldots, \mathbf{A}^q$ and $\mathbf{B}^1, \ldots, \mathbf{B}^p$. Using a single collection of sketches $\mathcal{S}(\mathbf{A}^1), \ldots, \mathcal{S}(\mathbf{A}^q), \mathcal{S}(\mathbf{B}^1), \ldots, \mathcal{S}(\mathbf{B}^p)$. Even if probabilites could be communicated, it is not clear what choice of probabilities should be used to make all pairs of sketches compatible.

## 1.3 OUR RESULTS

Our main contribution is to show that, despite the limitations above, sampling can in fact be applied effectively in the sketching setting by drawing on *coordinated random sampling* methods. Such methods include MinHash (Broder et al., 1998; Manasse et al., 2010; Ioffe, 2010), the $k$-minimum values (KMV) sketch (Beyer et al., 2007), conditional random sampling (Li et al., 2006), and coordinated variants of PPSWOR sampling (Cohen & Kaplan, 2007; 2013). We refer the reader to the recent survey of Cohen (2023) for a more complete review of prior work.

The idea behind coordinated sampling is to use a shared random seed to draw samples on two different machines that are likely to contain a significant number of shared indices, even if the exact same sampling probabilities are not used. In our setting, $\mathbf{A}$'s rows will be sampled using probabilities proportional to $\|\mathbf{A}_i\|_2^2$ and $\mathbf{B}$'s rows with probabilities proportional to $\|\mathbf{B}_i\|_2^2$. Our samples $\mathcal{S}(\mathbf{A})$ and $\mathcal{S}(\mathbf{B})$ are only likely to contain $\mathbf{A}_i$ and $\mathbf{B}_i$ for a shared index $i$ (which can then be used to estimate $\mathbf{A}^T\mathbf{B}$ via the sum $\mathbf{A}^T\mathbf{B} = \sum_{i=1}^n \mathbf{A}_i\mathbf{B}_i^T$) if *both* $\|\mathbf{A}_i\|_2^2$ and $\|\mathbf{B}_i\|_2^2$ are large. Fortunately, it turns out that this suffices to prove a bound equivalent to Fact 1.

Formally, we use a method for coordinated sampling without replacement called Priority Sampling (Ohlsson, 1998; Duffield et al., 2004; 2007) to prove the following main theoretical result:

**Theorem 2** (Main Result). *Consider* $\mathbf{A} \in \mathbb{R}^{n \times d}$, $\mathbf{B} \in \mathbb{R}^{n \times m}$, *and any* $\epsilon, \delta \in (0, 1)$. *There is a sketching procedure (Algorithm 1) that constructs sketches* $\mathcal{S}(\mathbf{A})$ *and* $\mathcal{S}(\mathbf{B})$ *consisting of at most* $k = \frac{2/\delta}{\epsilon^2} + 1$ *rows from* $\mathbf{A}$ *and* $\mathbf{B}$, *and there is a corresponding estimation procedure (Algorithm 2) that, using the information in these sketches, returns an estimate* $\mathbf{W}$ *such that, with probability* $1 - \delta$,

$$\|\mathbf{W} - \mathbf{A}^T\mathbf{B}\|_F \le \epsilon \|\mathbf{A}\|_F \|\mathbf{B}\|_F.$$

Theorem 2 matches the guarantee of the sampling method from Drineas et al. (2006a) up to a constant, albeit our method computes $\mathcal{S}(\mathbf{A})$ and $\mathcal{S}(\mathbf{B})$ completely independently from each other. As such, we match the state-of-the-art Fact 1 guarantee for linear sketching in the worst case, and improve on it whenever $\mathbf{A}$ and $\mathbf{B}$ are sparse. For example, in our experiments in Section 4, we consider some applications involving matrices with only 2% of entries in each row non-zero. For these applications, storing a subsample of size $k = O(1/\delta\epsilon^2)$ vs. a linear sketch $\Pi\mathbf{A}$ with height $k = O(1/\delta\epsilon^2)$ translates to a 50x savings in space.

Priority Sampling and related methods have recently been leveraged to give new sketching procedures for estimating *inner products*, which improve on linear sketching methods like Johnson-Lindenstrauss projection for that problem (Bessa et al., 2023; Daliri et al., 2024a;b). Inner products represent a special case of matrix products when $d = m = 1$. Our work significantly extends these results by addressing general matrix multiplication. In doing so, we encounter several technical challenges, including the fact that Priority Sampling—being a without-replacement sampling procedure—generates non-i.i.d. row samples from $\mathbf{A}$ and $\mathbf{B}$. We address these challenges in Section 2.

## 1.4 EXAMPLE APPLICATIONS

As mentioned, sketching methods are most useful in distributed computing environments, or in settings where we wish to compute many pairs of matrix-matrix products from a fixed collection of sketches.

**Multi-vector Retrieval.** One such setting arise in the "vector set search" or "multi-vector retrieval" problem, which has received recent attention (Engels et al., 2023; Dhulipala et al., 2024). This problem generalizes standard vector similarity search to matrices: we have a database of matrices $\mathbf{A}^1, \ldots, \mathbf{A}^q$ (each representing a document or media item) and another matrix $\mathbf{B}$ that represents a query. These matrices can be viewed as collections of column vectors, hence the name "vector set search". The goal is to find $\min_i d(\mathbf{A}^{i^T}\mathbf{B})$, where $d$ is some distance function that depends on the matrix-matrix product $\mathbf{A}^{i^T}\mathbf{B}$. Approximate matrix-multiplication can be used to speed up the computation of $\mathbf{A}^{i^T}\mathbf{B}$ and thus, the distance computation. For example, a Johnson-Lindenstrauss sketch is used in (Dhulipala et al., 2024). Here, the sketching setting is key: $\mathbf{A}^1, \ldots, \mathbf{A}^q$ are preprocessed into sketches that are computed before the query $\mathbf{B}$ is issued and $\mathcal{S}(\mathbf{B})$ cannot be chosen to depend on any particular $\mathbf{A}^i$ or $\mathcal{S}(\mathbf{A}^i)$, as it will be used to estimate $\mathbf{B}$'s matrix product with all $q$ matrices $\mathbf{A}^1, \ldots, \mathbf{A}^q$.

**Regression-based Dataset Search.** Another motivation of our work is to develop efficient methods for dataset search and discovery, a problem that has received significant interest in recent years Chepurko et al. (2020); Castelo et al. (2021); Liu et al. (2022); Ionescu et al. (2022).

In particular, suppose we have a data lake consisting of many datasets $\mathbf{A}^1, \ldots, \mathbf{A}^q$ and we want to support queries where a user provides a data vector $\mathbf{b}$ and the system returns all candidate datasets $\mathbf{A}^i$ that are *predictive* of $\mathbf{b}$. I.e., for which $\min_{\mathbf{x}} \|\mathbf{A}^i \mathbf{x} - \mathbf{b}\|_2$ is small. The sketching setting can be used to support such queries efficiently: $\mathbf{b}$ is sketched by the user and is sent to the dataset search system. It is then compared to precomputed sketches for each of $\mathbf{A}^1, \ldots, \mathbf{A}^q$ to approximate $\min_{\mathbf{x}} \|\mathbf{A}^i \mathbf{x} - \mathbf{b}\|_2$.

It turns out that this problem can be solved with a modified version of our matrix-product sketching method. Concretely, restricting our attention to a single $\mathbf{A} \in \mathbb{R}^{n \times d}$ and vector $\mathbf{b} \in \mathbb{R}^n$, our goal is to find $\tilde{\mathbf{x}}$ which is a near minimizer of the standard least squares problem: $\min_{\mathbf{x}} \|\mathbf{A}\mathbf{x} - \mathbf{b}\|_2^2$. The optimal $\mathbf{x}$ has the form $\mathbf{x}^* = (\mathbf{A}^T \mathbf{A})^{-1} \mathbf{A}^T \mathbf{b}$, where $\mathbf{A}^T \mathbf{A}$ is a relatively small, $d \times d$ matrix. So, the challenge is approximating the matrix-vector product $\mathbf{A}^T \mathbf{b}$ using compact sketches.

Approximating $\mathbf{A}^T \mathbf{b}$ directly using Theorem 2 does not suffice, as to ensure an accurate $\tilde{\mathbf{x}}$, we need small error with respect to a different norm than the standard Frobenius norm. Instead, we introduce a variant of Algorithm 1 that collects row samples from $\mathbf{A}$ based on the matrix's *statistical leverage scores*. Entries from $\mathbf{b}$ are sampled based on their squared magnitude. Our main result is as follows:

**Theorem 3** (Sketched Regression). *There is a procedure that constructs sketches $\mathcal{S}(\mathbf{A})$ and $\mathcal{S}(\mathbf{b})$ consisting of $O(d/\epsilon)$ row samples from $\mathbf{A} \in \mathbb{R}^{n \times d}$ and $\mathbf{b} \in \mathbb{R}^n$ such that, using only the information in those sketches, we can compute $\tilde{\mathbf{x}} \in \mathbb{R}^d$ satisfying, with probability at least $99/100$,*

$$\|\mathbf{A}\tilde{\mathbf{x}} - \mathbf{b}\|_2^2 \le \|\mathbf{A}\mathbf{x}^* - \mathbf{b}\|_2^2 + \epsilon \|\mathbf{b}\|_2^2.$$

Sketching algorithms for regression have been studied in prior work. For the problem above, the best existing result is based on linear sketching (e.g., Johnson-Lindenstrauss projection). Linear sketching methods achieve the same guarantee as Theorem 3 with a sketch of size $O(d^2/\epsilon)$ for $\mathbf{A}$ and a sketch of size $O(d/\epsilon)$ for $\mathbf{b}$ (specifically, the $d \times O(d/\epsilon)$ matrix $\mathbf{\Pi}\mathbf{A}$ and the vector $\mathbf{\Pi}\mathbf{b}$) Sarlós (2006); Woodruff (2014). Theorem 3 improves on these bounds when $\mathbf{A}$ is sparse. In particular, if $\mathbf{A}$ has $s \le d$ non-zeros per row, we require a sketch of size $O(sd/\epsilon)$ for $\mathbf{A}$ and of size $O(d/\epsilon)$ for $\mathbf{b}$.[1]

Theorem 3 is proven in Section 3. We remark that leverage score sampling has already been widely applied to regression problems outside of the distributed sketching setting (Drineas et al., 2006b; Cohen et al., 2015b; Chen & Price, 2019). It is well known that, if the rows of $\mathbf{A}$ and $\mathbf{b}$ are sampled with probability proportional to the leverage scores of $\mathbf{A}$, then a guarantee matching Theorem 3 holds as long as $O(d/\epsilon)$ samples are taken Sarlós (2006). However, standard leverage score sampling cannot be applied in our sketching setting since $\mathbf{A}$'s leverage scores cannot be used when subsampling $\mathbf{b}$. Again, this is not an issue with simply needing to communicate the scores. We want a sketch of $\mathbf{b}$ that is compatible with each matrix in a collection $\mathbf{A}^1, \ldots, \mathbf{A}^q$, which might have very different leverage score distributions. This challenge necessitates both a new algorithm and a new analysis.

Our work builds on a recent line of work that uses sketching methods for efficient dataset search in general. For example, sketching methods for estimating inner products have been applied to finding individual columns in a datalake that are highly correlated with a given query vector $\mathbf{b}$ (Santos et al., 2021; 2022; Daliri et al., 2024b). Our sketching methods for regression allow for more advanced search queries that go beyond pairwise correlation.

## 1.5 Notation and Preliminaries

Before proceeding, we briefly review notation used throughout the paper.

**Linear Algebra Notation.** The $i^{\text{th}}$ row of a matrix $\mathbf{A}$ is denoted by $\mathbf{A}_i$. The entry in the $i^{\text{th}}$ row and $j^{\text{th}}$ column of $\mathbf{A}$ is denoted by $\mathbf{A}_{i,j}$. For a vector $\mathbf{x}$, $\mathbf{x}_i$ denotes the $i^{\text{th}}$ entry. We use $\|\mathbf{x}\|_2$ to denote

---

[1]Linear sketching methods actually ensure a stronger guarantee: the additive error $\epsilon\|\mathbf{b}\|_2$ can be replaced with the residual $\epsilon\|\mathbf{A}\mathbf{x}^* - \mathbf{b}\|_2$, which is always smaller. In some applications, the difference is not significant. For example, in dataset search, most matrices $\mathbf{A}$ will be unrelated to $\mathbf{b}$, so we expect $\|\mathbf{A}\mathbf{x}^* - \mathbf{b}\|_2 \approx \|\mathbf{b}\|_2$. Additive error $\epsilon\|\mathbf{b}\|_2^2$ should suffice to at least rule out bad candidates. However a nice question for future work is to understand if more compact sketches can be obtained when targeting the stronger residual error guarantee.

the standard Euclidean norm of a vector $\mathbf{x}$ and $\|\mathbf{A}\|_F$ to denote the Frobenius norm of of matrix $\mathbf{A}$. The transpose of a matrix $\mathbf{A}$ is denoted by $\mathbf{A}^T$, and the inverse of $\mathbf{A}$ is denoted by $\mathbf{A}^{-1}$, provided it exists. The zero vector is denoted by $\mathbf{0}$, with dimension clear from context.

**Other Notation.** The expected value of a random variable $X$ is denoted by $\mathbb{E}[X]$, and its variance is denoted by $\mathrm{Var}(X)$. We use the notation $[n]$ to represent the set $\{1, \ldots, n\}$.

## 2 MATRIX PRODUCT SKETCHING WITH PRIORITY SAMPLING

Our main approach to matrix product sketching is based on subsampling. We can write any matrix product $\mathbf{A}^T\mathbf{B}$ as a sum of outer-products $\mathbf{A}^T\mathbf{B} = \sum_{i=1}^n \mathbf{A}_i\mathbf{B}_i^T$. We will estimate this sum as $\sum_{i \in \mathcal{T}} w_i\mathbf{A}_i\mathbf{B}_i^T$ where $\mathcal{T}$ is a small subset of $\{1, \ldots, n\}$ and $w_i$ is an appropriately chosen weight. Typically $\mathcal{T}$ is selected via importance sampling: indices $i$ that correspond to larger norm rows in $\mathbf{A}$ or $\mathbf{B}$ are sampled with higher probability (Drineas et al., 2006a). The challenge in the sketching setting is that $\mathbf{A}$ and $\mathbf{B}$ must be sampled independently from each other, without knowledge of the other matrices row norms.

We address this issue by using a coordinated sampling technique known as Priority Sampling, which has been widely used for subsampling data streams Duffield et al. (2007), and more recently for subsampling vectors for inner product estimation (Daliri et al., 2024b). Pseudocode for the method is included in Algorithm 1. To give better intuition for the method, we informally describe another closely related algorithm called Threshold Sampling, which gives the same guarantees as Priority Sampling for our problem, but has the disadvantage of producing a sketch whose size can only be bounded in expectation.

Threshold Sampling works as follows: 1) using shared random bits, we select a random hash function $h : \{1, \ldots, n\} \to [0, 1]$ that assigns a uniformly random number between $[0, 1]$ to any index $i$.[2], 2) we collect in the sketch $\mathcal{S}(\mathbf{A})$ any row $\mathbf{A}_i$ for which $h(i) \leq k \cdot \|\mathbf{A}_i\|_2^2/\|\mathbf{A}\|_F^2$, and in the sketch $\mathcal{S}(\mathbf{B})$ any row $\mathbf{b}_i$ for which $h(i) \leq k \cdot \|\mathbf{B}_i\|_2^2/\|\mathbf{B}\|_F^2$. Equivalently, $\mathbf{A}_i$ is sampled if the reweighted hash value $h(i)/\|\mathbf{A}_i\|_2^2$ falls below a fixed *threshold* $k/\|\mathbf{A}\|_F^2$ (and likewise for $\mathbf{B}_i$).

It is easy to see that each sketch contains $k$ rows in expectation. Moreover, since we use a shared hash function, it can be checked that, for any index $i$, we have that *both* $\mathbf{A}_i \in \mathcal{S}(\mathbf{A})$ and $\mathbf{B}_i \in \mathcal{S}(\mathbf{B})$ with probability:

$$p_i = \min\left(1, k \cdot \|\mathbf{A}_i\|_2^2/\|\mathbf{A}\|_F^2, k \cdot \|\mathbf{B}_i\|_2^2/\|\mathbf{B}\|_F^2\right).$$

Let $\mathcal{T}$ denote the set of indices that appear in both sketches. We return the unbiased estimate $\mathbf{W} = \sum_{i \in \mathcal{T}} \frac{1}{p_i}\mathbf{A}_i\mathbf{B}_i^T$. To show that this estimate is accurate, we can follow an analysis similar to the original paper on subsampled randomized matrix multiplication, which bounds the expected squared error $\mathbb{E}\|\mathbf{W} - \mathbf{A}^T\mathbf{B}\|_F^2$ before applying Markov's inequality (Drineas et al., 2006a). The only difference is that we must show that it suffices to sample indices with probability proportional to the *minimum* of $\|\mathbf{A}_i\|_2^2/\|\mathbf{A}\|_F^2$ and $\|\mathbf{B}_i\|_2^2/\|\mathbf{B}\|_F^2$ instead of the product of these numbers. Perhaps the fact that this suffices is intuitive: for the outerproduct $\mathbf{A}_i\mathbf{B}_i^T$ to make a significant contribution to $\mathbf{A}^T\mathbf{B}$, neither $\mathbf{A}_i$ nor $\mathbf{B}_i$ can have small magnitude. A full analysis of Threshold Sampling is given in Appendix B.

The method we propose, Priority Sampling, is almost identical to Threshold Sampling. However, instead of fixing the threshold $k/\|\mathbf{A}\|_F^2$, which leads to a random number of indices being sampled, we *dynamically* set the threshold to collect exactly $k$ samples. Doing so does not change the method in spirit, but complicates the analysis since samples are no longer independent.

Nevertheless, drawing inspiration from a new, simple analysis of Priority Sampling for sampling numbers from a stream (Daliri et al., 2024a), we are able to prove the following bound:

**Theorem 4.** *Let* $\mathbf{A} \in \mathbb{R}^{n \times d}$, $\mathbf{B} \in \mathbb{R}^{n \times m}$, *and let* $\mathcal{S}(\mathbf{A}) = \{\mathcal{I}_\mathbf{A}, V_\mathbf{A}, \tau_\mathbf{A}\}$ *and* $\mathcal{S}(\mathbf{B}) = \{\mathcal{I}_\mathbf{B}, V_\mathbf{B}, \tau_\mathbf{B}\}$ *be sketches produced by Algorithm 1 with input* $k$ *and a shared seed* $s$. *Suppose* $\mathbf{W}$ *is the approximate*

---

[2]In practice, $h$ can be substituted with a pseudorandom function mapping to a large discrete subset of $[0, 1]$. For simplicity, we assume access to a real-valued, perfect hash function, as is standard in the literature (Cormode et al., 2011)

---

**Algorithm 1** Priority Sampling

---

**Input:** Matrix $\mathbf{A}$ of size $n \times d$, random seed $s$, number of row samples, $k$.
**Output:** Sketch $\mathcal{S}(\mathbf{A}) = \{\mathcal{I}_\mathbf{A}, V_\mathbf{A}, \tau_\mathbf{A}\}$, where $\mathcal{I}_\mathbf{A}$ is a subset of row indices from $\{1, \ldots, n\}$ and $V_\mathbf{A}$ contains $\mathbf{A}_i$ for all $i \in \mathcal{I}_\mathbf{A}$, $\tau_\mathbf{A}$ is the threshold used to determine whether a row in $\mathcal{I}_\mathbf{A}$ is selected

---

1: Use random seed $s$ to select a uniformly random hash function $h : \{1, ..., n\} \to [0, 1]$.
2: Initialize $\mathcal{I}_\mathbf{A}$ and $V_\mathbf{A}$ to be empty lists.
3: Compute rank $R_i = \frac{h(i)}{\|\mathbf{A}_i\|_2^2}$ for all $i$ such that $\mathbf{A}_i \neq \mathbf{0}$.
4: Set $\tau_\mathbf{A}$ equal to the $(k+1)^{\text{st}}$ smallest value $R_i$, or set $\tau_\mathbf{A} = \infty$ if $\mathbf{A}$ has $< k + 1$ non-zero rows.
5: **for** $i$ such that $\mathbf{A}_i \neq \mathbf{0}$ **do**
6:     **if** $R_i < \tau_\mathbf{A}$ **then**
7:         Append $i$ to $\mathcal{I}_\mathbf{A}$, append $\mathbf{A}_i$ to $V_\mathbf{A}$.
8: **return** $\mathcal{S}(\mathbf{A}) = \{\mathcal{I}_\mathbf{A}, V_\mathbf{A}, \tau_\mathbf{A}\}$

---

**Algorithm 2** Approximate Matrix Multiplication

---

**Input:** Sketches $\mathcal{S}(\mathbf{A}) = \{\mathcal{I}_\mathbf{A}, V_\mathbf{A}, \tau_\mathbf{A}\}$, $\mathcal{S}(\mathbf{B}) = \{\mathcal{I}_\mathbf{B}, V_\mathbf{B}, \tau_\mathbf{B}\}$ constructed by Algorithm 1.
**Output:** Estimate $\mathbf{W}$ for $\mathbf{A}^T \mathbf{B}$.

---

1: Compute $\mathcal{T} = \mathcal{I}_\mathbf{A} \cap \mathcal{I}_\mathbf{B}$. Note that for all $i \in \mathcal{T}$, $V_\mathbf{A}$ and $V_\mathbf{B}$ contain $\mathbf{A}_i$ and $\mathbf{B}_i$.
2: **return**
$$\mathbf{W} = \sum_{i \in \mathcal{T}} \frac{\mathbf{A}_i \mathbf{B}_i^T}{\min(1, \|\mathbf{A}_i\|_2^2 \cdot \tau_\mathbf{A}, \|\mathbf{B}_i\|_2^2 \cdot \tau_\mathbf{B})}.$$

---

*matrix of $\mathbf{A}^T \mathbf{B}$ calculated using Algorithm 2 on these sketches. Then, $\mathbb{E}[\mathbf{W}] = \mathbf{A}^T \mathbf{B}$ and*

$$\mathbb{E}\left[\|\mathbf{W} - \mathbf{A}^T \mathbf{B}\|_F^2\right] \leq \frac{2}{k-1} \|\mathbf{A}\|_F^2 \|\mathbf{B}\|_F^2.$$

*Additionally, $|\mathcal{I}_\mathbf{A}| \leq k$ and $|\mathcal{I}_\mathbf{B}| \leq k$. I.e., each sketch contains no more than $k$ rows from $\mathbf{A}$ and $\mathbf{B}$, respectively. If each matrix has at least $k$ non-zero rows, we have that $|\mathcal{I}_\mathbf{A}| = |\mathcal{I}_\mathbf{B}| = k$.*

We prove Theorem 4 via Lemma 5, which is proven in Appendix A due to space limitations.

**Lemma 5.** *Let $\mathbf{A}, \mathbf{B}$, and $\mathbf{W}$ be as in Theorem 4. For any $x, y \in [d] \times [m]$ we have:*

$$\mathbb{E}[\mathbf{W}_{x,y}] = [\mathbf{A}^T \mathbf{B}]_{x,y} \text{ and } \mathbb{E}\left[(\mathbf{W}_{x,y} - [\mathbf{A}^T \mathbf{B}]_{x,y})^2\right] \leq \sum_{i=1}^n \frac{\mathbf{A}_{i,x}^2 \mathbf{B}_{i,y}^2}{k-1} \left(\frac{\|\mathbf{A}\|_F^2}{\|\mathbf{A}_i\|_2^2} + \frac{\|\mathbf{B}\|_F^2}{\|\mathbf{B}_i\|_2^2}\right).$$

Lemma 5 gives an entrywise guarantee on the error of the approximation $\mathbf{W}$, which we can then use to give an overall bound on the Frobenius norm error. That analysis is given below.

*Proof of Theorem 4.* The entrywise expectation guarantee of Lemma 5 immediately gives $\mathbb{E}[\mathbf{W}] = \mathbf{A}^T \mathbf{B}$. We are left to bound the expected Frobenius error, which can be written as a sum over entries:

$$
\begin{aligned}
\mathbb{E}\left[\|\mathbf{W} - \mathbf{A}^T \mathbf{B}\|_F^2\right] &= \sum_{x,y} \mathbb{E}\left[\left(\mathbf{W}_{x,y} - [\mathbf{A}^T \mathbf{B}]_{x,y}\right)^2\right] \leq \sum_{x,y} \sum_{i=1}^n \frac{\mathbf{A}_{i,x}^2 \mathbf{B}_{i,x}^2}{k-1} \left(\frac{\|\mathbf{A}\|_F^2}{\|\mathbf{A}_i\|_2^2} + \frac{\|\mathbf{B}\|_F^2}{\|\mathbf{B}_i\|_2^2}\right) \\
&= \frac{1}{k-1} \sum_{i=1}^n \left(\frac{\|\mathbf{A}\|_F^2}{\|\mathbf{A}_i\|_2^2} + \frac{\|\mathbf{B}\|_F^2}{\|\mathbf{B}_i\|_2^2}\right) \sum_{x=1}^d \mathbf{A}_{i,x}^2 \sum_{y=1}^m \mathbf{B}_{i,y}^2 \\
&= \frac{1}{k-1} \sum_{i=1}^n \left(\frac{\|\mathbf{A}\|_F^2}{\|\mathbf{A}_i\|_2^2} + \frac{\|\mathbf{B}\|_F^2}{\|\mathbf{B}_i\|_2^2}\right) \|\mathbf{A}_i\|_2^2 \|\mathbf{B}_i\|_2^2 \\
&= \frac{1}{k-1} \sum_{i=1}^n \|\mathbf{A}\|_F^2 \|\mathbf{B}_i\|_2^2 + \|\mathbf{B}\|_F^2 \|\mathbf{A_i}\|_2^2 = \frac{2}{k-1} \|\mathbf{A}\|_F^2 \|\mathbf{B}\|_F^2. \qquad \square
\end{aligned}
$$

With Theorem 4 in place, our main result, Theorem 2 follows as an immediate corollary:

*Proof of Theorem 2.* Our main result, Theorem 2, follows as an immediate corollary of Theorem 4. In particular, if we set $k = \frac{2/\delta}{\epsilon^2} + 1$, then we have that $\mathbb{E}\left[\|\mathbf{W} - \mathbf{A}^T\mathbf{B}\|_F^2\right] \leq \epsilon^2\delta\|\mathbf{A}\|_F^2\|\mathbf{B}\|_F^2$. Applying Markov's inequality proves the theorem. $\qquad\square$

## 3 SKETCHED REGRESSION

In this section, we focus on proving Theorem 3. To do so, we first prove a simpler version of the result that, instead of just storing a subsample of rows from $\mathbf{A}$ in $\mathcal{S}(\mathbf{A})$, also explicitly stores the $d \times d$ covariance matrix $\mathbf{A}^T\mathbf{A}$. The pseudocode for this method is included as Algorithm 3. It leads to a sketch of size $O(d^2 + ds/\epsilon)$ when $\mathbf{A}$ has $s$ non-zeros per row. We later show that the sketch can be modified to have size $O(ds/\epsilon)$ by replacing $\mathbf{A}^T\mathbf{A}$ with another subsample of rows from $\mathbf{A}$.

*Proof of Theorem 3.* Our goal is to compute a vector $\tilde{\mathbf{x}}$ that approximates $\mathbf{x}^* = (\mathbf{A}^T\mathbf{A})^{-1}\mathbf{A}^T\mathbf{b}$. In particular, we wish to obtain an upper bound on $\|\mathbf{A}\tilde{\mathbf{x}} - \mathbf{b}\|_2^2$. Observing that $\mathbf{A}\mathbf{x}^* - \mathbf{b}$ is orthogonal to any vector in the column span of $\mathbf{A}$, we can apply Pythagorean theorem to write:

$$\|\mathbf{A}\tilde{\mathbf{x}} - \mathbf{b}\|_2^2 = \|\mathbf{A}\mathbf{x}^* - \mathbf{b}\|_2^2 + \|\mathbf{A}\tilde{\mathbf{x}} - \mathbf{A}\mathbf{x}^*\|_2^2,$$

or equivalently:

$$\|\mathbf{A}\tilde{\mathbf{x}} - \mathbf{b}\|_2^2 - \|\mathbf{A}\mathbf{x}^* - \mathbf{b}\|_2^2 = \|\mathbf{A}\tilde{\mathbf{x}} - \mathbf{A}(\mathbf{A}^T\mathbf{A})^{-1}\mathbf{A}^T\mathbf{b}\|_2^2. \tag{1}$$

We claim that the sketching and regression procedure in Algorithm 3 returns $\tilde{\mathbf{x}}$ such that $\mathbf{A}\tilde{\mathbf{x}}$ is exactly equal to $\mathcal{F}(\mathcal{S}(\mathbf{A}(\mathbf{A}^T\mathbf{A})^{-1}\mathbf{A}^T), \mathcal{S}(\mathbf{b}))$, where $\mathcal{S}(\cdot)$ denotes the sketching procedure of Algorithm 1 and $\mathcal{F}(\cdot)$ denotes the estimation procedure of Algorithm 2. However, it does so in an implicit way, without every explicitly forming the large $n \times n$ and possibly dense matrix $\mathbf{A}(\mathbf{A}^T\mathbf{A})^{-1}\mathbf{A}^T$. If this claim holds, then the main guarantee of Theorem 3 immediately follows. In particular, by the guarantee of Theorem 2, as long as we choose sketch size $k = O(d/\epsilon)$, we would have:

$$\|\mathbf{A}\tilde{\mathbf{x}} - \mathbf{A}(\mathbf{A}^T\mathbf{A})^{-1}\mathbf{A}^T\mathbf{b}\|_2^2 \leq \frac{\epsilon}{d}\|\mathbf{A}(\mathbf{A}^T\mathbf{A})^{-1}\mathbf{A}^T\|_F^2\|\mathbf{b}\|_2^2 = \frac{\epsilon}{d} \cdot d\|\mathbf{b}\|_2^2 = \epsilon\|\mathbf{b}\|_2^2.$$

Above we have used that $\|\mathbf{A}(\mathbf{A}^T\mathbf{A})^{-1}\mathbf{A}^T\|_F^2 = \text{tr}(\mathbf{A}(\mathbf{A}^T\mathbf{A})^{-1}\mathbf{A}^T\mathbf{A}(\mathbf{A}^T\mathbf{A})^{-1}\mathbf{A}^T) = \text{tr}(\mathbf{I}_d) = d$. Plugging into equation 1 would then prove the theorem.

So, it is left to establish that Algorithm 3 returns $\tilde{\mathbf{x}}$ such that $\mathbf{A}\tilde{\mathbf{x}}$ is exactly equal to $\mathcal{F}(\mathcal{S}(\mathbf{A}(\mathbf{A}^T\mathbf{A})^{-1}\mathbf{A}^T), \mathcal{S}(\mathbf{b}))$. For this to be the case, it can be checked that it suffices to simply sample from $\mathbf{A}$ with probabilities proportional to the squared row norms in $\mathbf{A}(\mathbf{A}^T\mathbf{A})^{-1}\mathbf{A}^T$. Then, multiplying the sampled rows by $(\mathbf{A}^T\mathbf{A})^{-1}$ to produce $\tilde{\mathbf{x}}$, and again by $\mathbf{A}$ to produce $\mathbf{A}\tilde{\mathbf{x}}$ exactly reproduces $\mathcal{F}(\mathcal{S}(\mathbf{A}(\mathbf{A}^T\mathbf{A})^{-1}\mathbf{A}^T), \mathcal{S}(\mathbf{b}))$.

The $i^{\text{th}}$ squared row norm of $\mathbf{A}(\mathbf{A}^T\mathbf{A})^{-1}\mathbf{A}^T$ can be written as $\|\mathbf{A}(\mathbf{A}^T\mathbf{A})^{-1}\mathbf{A}^T\mathbf{e}_i\|_2^2$, where $\mathbf{e}_i$ denotes the $i^{\text{th}}$ standard basis vector. We then have:

$$\|\mathbf{A}(\mathbf{A}^T\mathbf{A})^{-1}\mathbf{A}^T\mathbf{e}_i\|_2^2 = \mathbf{e}_i^T\mathbf{A}(\mathbf{A}^T\mathbf{A})^{-1}\mathbf{A}^T\mathbf{A}(\mathbf{A}^T\mathbf{A})^{-1}\mathbf{A}^T\mathbf{e}_i = \mathbf{A}_i^T(\mathbf{A}^T\mathbf{A})^{-1}\mathbf{A}_i.$$

The quantity above is exactly equal to the $i^{\text{th}}$ *statistical leverage score* of $\mathbf{A}$, which is the quantity that Theorem 3 uses when Priority Sampling, so the claim holds. We note that, besides the naive approach, efficient algorithms are known for more quickly computing the statistical leverage scores of a matrix $\mathbf{A}$, although our focus here is on sketch size as opposed to construction time (Mahoney et al., 2012; Clarkson & Woodruff, 2013)

**Optimized Method.** The approach above immediately gives an $O(d^2 + ds/\epsilon)$ space sketch for least squares regression when $\mathbf{A}$ has $s$-sparse rows. This already improves on the space complexity of $O(d^2/\epsilon)$ achieved by linear sketching methods. However, in settings where $d$ is large, it would be better to avoid a quadratic dependence on $d$ entirely. To do so, instead of explicitly storing the $d \times d$ matrix $\mathbf{A}^T\mathbf{A}$ in our sketch, we can store $\mathbf{SA}$, where $\mathbf{S} \in \mathbb{R}^{z \times d}$ is a matrix that selects and reweights $z$ rows from $\mathbf{A}$. Instead of returning $\tilde{\mathbf{x}} = (\mathbf{A}^T\mathbf{A})^{-1}\mathbf{W}$ as in Line 6 of Algorithm 3, we would return:

$$\tilde{\mathbf{x}}' = (\mathbf{A}^T\mathbf{S}^T\mathbf{SA})^{-1}\mathbf{W}.$$

---

**Algorithm 3** Sketching for Regression (not optimized)

---

**Input:** Matrix $\mathbf{A}_{n \times d}$, matrix $\mathbf{b}_{n \times 1}$, randomness seed $s$, and target error $\epsilon$.

1: Compute $\ell_i$ as the leverage score of $\mathbf{A}$: $\ell_i = \mathbf{A}_i(\mathbf{A}^T\mathbf{A})^{-1}\mathbf{A}_i^T$.

2: Construct sketches $\mathcal{S}(\mathbf{A})$ with a target sampling size of $O\left(\frac{d}{\epsilon}\right)$ (rows) using Algorithm 1 with a shared seed $s$. However, compute the rank (line 2 of Algorithm 1) as $R_i = \frac{h(i)}{\ell_i}$.

3: Construct sketches $\mathcal{S}(\mathbf{b})$ with a target sampling size of $O\left(\frac{d}{\epsilon}\right)$ (rows) using Algorithm 1 with a shared seed $s$.

**Output:** $\left(\mathcal{S}(\mathbf{A}), \mathbf{A}^T\mathbf{A}\right), \mathcal{S}(\mathbf{b})$

---

**Procedure** REGRESSION $\left((\mathbf{A}^T\mathbf{A}, \mathcal{S}(\mathbf{A}), \mathcal{S}(\mathbf{b})\right)$.

4: Compute Compute $\ell_i$ as the leverage score of $\mathbf{A}$: $\ell_i = \mathbf{A}_i(\mathbf{A}^T\mathbf{A})^{-1}\mathbf{A}_i^T$ for any $\mathbf{A}_i$ in $\mathcal{S}(\mathbf{A})$.

5: Compute $\mathcal{T} = \mathcal{I}_\mathbf{A} \cap \mathcal{I}_\mathbf{b}$.

6: Compute $\mathbf{W} = \sum_{i \in \mathcal{T}} \frac{\mathbf{A}_i \mathbf{b}_i}{\min(1, \ell_i \cdot \tau_\mathbf{A}, \mathbf{b}_i^2 \cdot \tau_\mathbf{b})}$.

**Output:** $\tilde{\mathbf{x}} = (\mathbf{A}^T\mathbf{A})^{-1}\mathbf{W}$

---

It is well known that there exist choices of $\mathbf{S}$ with $m = O(d/\epsilon)$ rows such that $\mathbf{A}^T\mathbf{S}^T\mathbf{S}\mathbf{A}$ is a $\sqrt{\epsilon}$-relative error spectral approximation to $\mathbf{A}^T\mathbf{A}$ (Batson et al., 2012). I.e.,

$$(1 - \sqrt{\epsilon})\mathbf{A}^T\mathbf{A} \preceq \mathbf{A}^T\mathbf{S}^T\mathbf{S}\mathbf{A} \preceq (1 + \sqrt{\epsilon})\mathbf{A}^T\mathbf{A} \text{ and}$$

$$(1 - \sqrt{\epsilon})(\mathbf{A}^T\mathbf{A})^{-1} \preceq (\mathbf{A}^T\mathbf{S}^T\mathbf{S}\mathbf{A})^{-1} \preceq (1 + \sqrt{\epsilon})(\mathbf{A}^T\mathbf{A})^{-1},$$

where $\preceq$ denotes the Loewner order. Given the second guarantee, we can check that $\|\mathbf{A}(\mathbf{A}^T\mathbf{A})^{-1}\mathbf{A}^T - \mathbf{A}(\mathbf{A}^T\mathbf{S}^T\mathbf{S}\mathbf{A})^{-1}\mathbf{A}^T\|_2 \leq \sqrt{\epsilon}$, where $\| \cdot \|_2$ denotes the operator norm.

Now, observe that $\mathbf{A}\tilde{\mathbf{x}}' = \mathbf{A}(\mathbf{A}^T\mathbf{S}^T\mathbf{S}\mathbf{A})^{-1}\mathbf{A}^T\mathbf{A}\tilde{\mathbf{x}}$, and thus:

$$\|\mathbf{A}\tilde{\mathbf{x}}' - \mathbf{A}\tilde{\mathbf{x}}\|_2 = \|\mathbf{A}(\mathbf{A}^T\mathbf{S}^T\mathbf{S}\mathbf{A})^{-1}\mathbf{A}^T\mathbf{A}\tilde{\mathbf{x}} - \mathbf{A}(\mathbf{A}^T\mathbf{A})^{-1}\mathbf{A}^T\mathbf{A}\tilde{\mathbf{x}}\|_2 \leq \sqrt{\epsilon}\|\mathbf{A}\tilde{\mathbf{x}}\|_2 \leq 2\sqrt{\epsilon}\|\mathbf{b}\|_2.$$

In the last step, we used that $\|\mathbf{A}\tilde{\mathbf{x}} - \mathbf{b}\|_2 \leq \epsilon\|\mathbf{b}\|_2$ to loosely upper bound $\|\mathbf{A}\tilde{\mathbf{x}}\|_2$. Finally, we put everything together. As proven earlier, $\|\mathbf{A}\tilde{\mathbf{x}} - \mathbf{A}\mathbf{x}^*\|_2 \leq \sqrt{\epsilon}\|\mathbf{b}\|_2$. Applying triangle inequality, we thus that that $\|\mathbf{A}\tilde{\mathbf{x}}' - \mathbf{A}\mathbf{x}^*\|_2 \leq 3\sqrt{\epsilon}\|\mathbf{b}\|_2$. Applying Pythagorean theorem as before, we conclude that $\|\mathbf{A}\tilde{\mathbf{x}}' - \mathbf{b}\|_2^2 \leq \|\mathbf{A}\mathbf{x}^* - \mathbf{b}\|_2^2 + 9\epsilon\|\mathbf{b}\|_2^2$. Adjusting $\epsilon$ by a constant gives the desired result. $\square$

We remark that one way of producing a matrix $\mathbf{S}$ satisfying the spectral approximation guarantee above is to subsample and reweight $m = O(d \log d/\epsilon)$ rows from $\mathbf{A}$ by leverage scores. While worse by a $\log d$ factor than the deterministic methods given e.g. in Batson et al. (2012), the advantage of such an approach is that we can *reuse* the samples used to approximate $\mathbf{A}^T\mathbf{A}$ when approximating $\mathbf{A}^T\mathbf{b}$, for which we also sample via leverage scores. This "single sketch" procedure is what we implement in our experimental section.

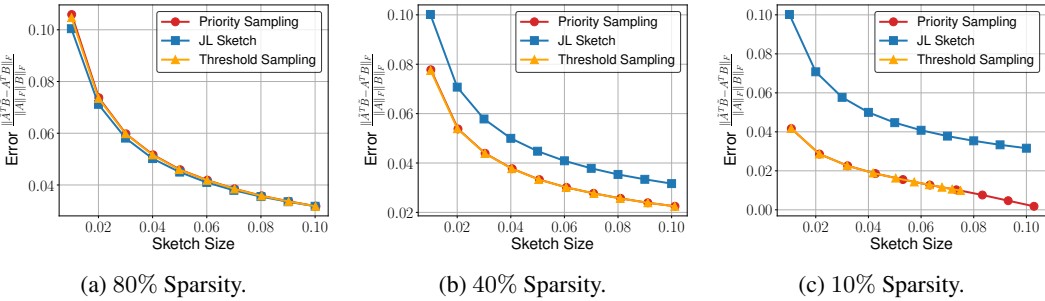

    (a) 80% Sparsity.         (b) 40% Sparsity.         (c) 10% Sparsity.

Figure 1: Performance of matrix product sketching over synthetic data with varying sparsity levels (10%, 40%, and 80%). Priority sampling and threshold sampling are depicted on top of each other and both methods outperform the JL sketch as the level of sparsity increases.

## 4 EXPERIMENTS

We experimentally evaluate our sampling-based matrix product sketches in a variety of settings. First, we use synthetic data to directly compare the method to the standard linear sketching approach:

Johnson-Lindenstrauss random project (which we denote as JL Sketch in our plots). As predicted by our theory, the methods perform similarly for dense matrices, but our Priority Sampling method obtains more compact sketches for sparse matrices. We also apply our method to a number of real-world problems, including to regression tasks, as outlined in Section 3. We show that for two popular datasets, our method can outperform over the best existing linear sketch.

Additionally, we apply our Priority Sampling method to approximating matrix multiplications that arise in transformer-based large language models. Deploying auto regressive language models involves performing attention decoding in an online setting, where key and value embeddings from each transformer layer are cached to eliminate redundant computations. More precisely, during each token generation phase, the stream of tokens is encapsulated by three matrices known as the query ($\mathbf{Q}$), key ($\mathbf{K}$), and value ($\mathbf{V}$) embeddings. At the heart of this process, each iteration involves calculating the attention matrix as $\texttt{Att} = \texttt{Softmax}(\mathbf{Q}\mathbf{K}^T/\sqrt{d})\mathbf{V}$ where $d$ is the embedding dimension. Recent studies (Hooper et al., 2024; Zirui Liu et al., 2023) have introduced methods that apply vector quantization to the key and value embeddings, replacing the full matrices with a quantized matrix. In this study, we employ Priority Sampling to sketch $\mathbf{Q}$ and $\mathbf{K}$. We assess the performance of our approach in approximating $\mathbf{Q}\mathbf{K}^T$ compared to linear sketching techniques (see appendix C for detailed results). Moreover, recent work Zandieh et al. (2024) introduced a quantized JL-based method for KV cache compression, outperforming previous baselines. However, our results show that Priority Sampling achieves even greater accuracy and efficiency, highlighting its potential for matrix product sketching in KV cache sketching.

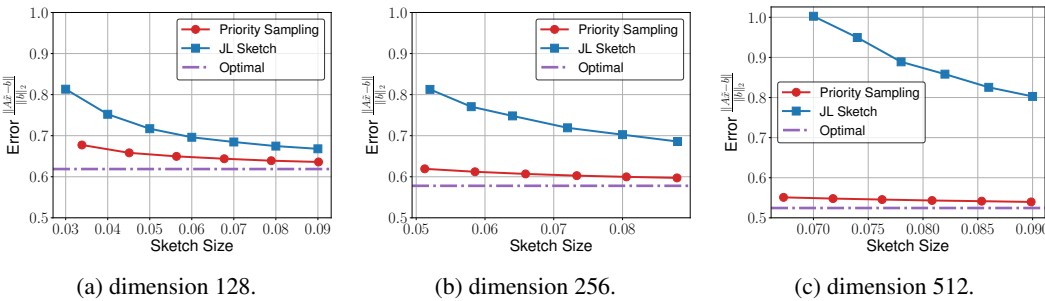

(a) dimension 128.    (b) dimension 256.    (c) dimension 512.

Figure 2: Comparison of Regression Sketching Methods on the IMDB Dataset: The plots illustrate the approximation error of different sketching methods across various sketch sizes. The matrix $\mathbf{A}$ is generated using TF-IDF on 10,000 random reviews, keeping the top 256, 512, and 1024 features. As the dimensionality increases, the matrices become more sparse. The matrix $\mathbf{b}$ represents the sentiment scores (positivity or negativity) of the reviews.

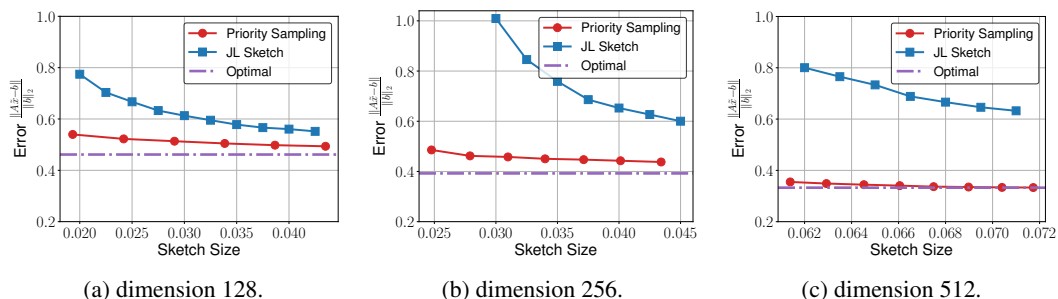

(a) dimension 128.    (b) dimension 256.    (c) dimension 512.

Figure 3: Sketched Regression Methods on the Android Review Dataset: The plots illustrate the approximation error of different sketching methods across various sketch sizes. The matrix $\mathbf{A}$ is generated using sparse transformer SPLADE (Formal et al., 2022) over 10,000 random reviews, retaining the top 128, 256, and 512 important features. The matrix $\mathbf{b}$ represents the review scores.

**Datasets** For the task of sketched regression, we use two primary datasets. The first dataset includes Android application reviews with user ratings, which reflect the quality of the applications (Grano et al., 2017). The second dataset contains IMDB movie reviews, labeled as positive or negative (Maas et al., 2011).

We transform the reviews into sparse vector embeddings using TF-IDF and SPLADE (Formal et al., 2022). Regression analysis is then performed to explore the relationship between the vectorized reviews and their ratings (Figure 3) or sentiment labels (Figure 2).

Additionally, we evaluate our model on synthetic datasets (Figure 1) for the task of approximate matrix multiplication $\mathbf{A}^T\mathbf{B}$. The entries of the matrices $\mathbf{A}$ and $\mathbf{B}$ are generated from a Gaussian distribution $N(0, 1)$. However, 10% of the dataset includes outliers, with values that are 10 times higher. The choice of 10% outliers reflects a moderate level of noise typically observed in real-world datasets, where outliers, while not dominating the data, can still have a substantial impact on the outcome. Testing with outliers allows us to assess the robustness of our model under such conditions. To examine how varying the sparsity of the matrices $\mathbf{A}$ and $\mathbf{B}$ affects approximation, we modify the number of non-zero entries. Both matrices are flattened, and we keep only a designated percentage of entries non-zero.

Alongside these datasets, we use the (Bai et al., 2023) dataset to produce a long text prompt from its `MultiFieldQA` dataset for the task of KV cache in transformers. We compare our sketch as a quantization method for the Key to reduce cache usage (Figure 4).

**Sketching Size** For sampling-based sketches, it is necessary to store the indices of the sampled rows. We account for both the size of each index and the need to store selected items in full precision. In contrast, for linear sketches, only the output of the projected matrix $\mathbf{\Pi A}$ needs to be stored. We report the total number of bits required for storage across all approaches and present the relative size of the sketches compared to the original matrices in bits, referred to as the compression ratio. Additionally, for threshold sampling, the sketch size is not fixed due to the nature of the algorithm. Therefore, instead of reporting the expected value, we provide the average compression ratio.

**Estimation Error** For the plots of matrix multiplication $\mathbf{A}$ and $\mathbf{B}$, we report the absolute difference between the estimated product and the true product, divided by $\|\mathbf{A}\|_F\|\mathbf{B}\|_F$. As stated in theorem 2, this term appears on the right-hand side of the accuracy guarantee for approximate matrix multiplication. For all plots regarding the regression between $\mathbf{A}$ and $\mathbf{b}$, we report the absolute difference between the estimated value and the true value, divided by $\|\mathbf{b}\|_2$.

**Interpretation of Experimental Results** As we can see, for both the tasks of matrix multiplication (Figure 1, Figure 4) and sketched regression (Figure 2, Figure 3), we have observed that as the matrices become sparser, our method improves over the best-known linear sketching methods. Even though our method needs to allocate some of its budget to store indices, it still outperforms JL methods.

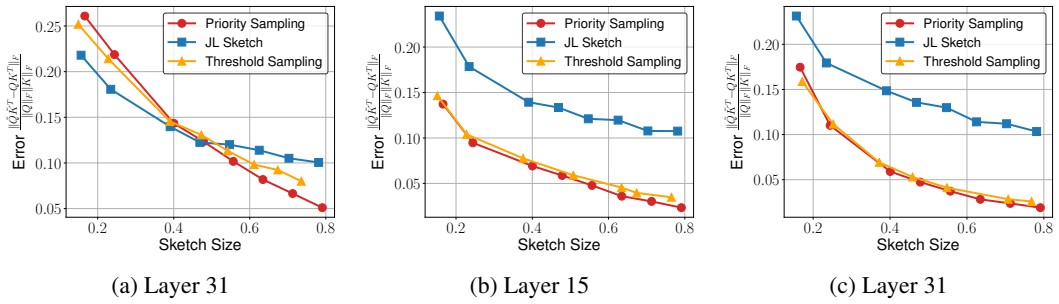

(a) Layer 31          (b) Layer 15          (c) Layer 31

Figure 4: Comparison of KV Cache Sketching Methods on the LongBench for `MultiFieldQA`: The plots show the accuracy of different sketching methods approximating $\mathbf{QK}^T$ across various sketch sizes. The matrices $\mathbf{Q}$ and $\mathbf{K}$ are generated from prompt tokens, and the approximation errors are displayed. Layers refer to the individual layers (hidden layers) of the Transformer architecture in the LLaMA 2 model (`meta-llama/Llama-2-7b-chat-hf`).

## 5 ACKNOWLEDGMENTS

This work was supported by NSF Award IIS-2106888.

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

## A   SUPPORTING PROOFS

We begin by proving Lemma 5, which was the key result used in proving our main matrix product sketching result.

*Proof of Lemma 5.* For any $x, y \in [d] \times [m]$ we can write the $x, y$ entry of our estimate $\mathbf{W}$ as:

$$\mathbf{W}_{x,y} = \sum_{i \in \mathcal{T}} \frac{\mathbf{A}_{i,x}\mathbf{B}_{i,y}}{\min(1, \|\mathbf{A}_i\|_2^2 \cdot \tau_{\mathbf{A}}, \|\mathbf{B}_i\|_2^2 \cdot \tau_{\mathbf{B}})}.$$

Recall that we aim to prove the following:

$$\mathbb{E}\left[\mathbf{W}_{x,y}\right] = \left[\mathbf{A}^T\mathbf{B}\right]_{x,y} \text{ and } \mathbb{E}[(\mathbf{W}_{x,y} - \left[\mathbf{A}^T\mathbf{B}\right]_{x,y})^2] \leq \sum_{i=1}^{n} \frac{\mathbf{A}_{i,x}^2\mathbf{B}_{i,y}^2}{k-1}\left(\frac{\|\mathbf{A}\|_F^2}{\|\mathbf{A}_i\|_2^2} + \frac{\|\mathbf{B}\|_F^2}{\|\mathbf{B}_i\|_2^2}\right).$$

For any $i$, let $\mathbb{1}$ be a 0-1 indicator random variable for the event that index $i$ is selected for both $\mathcal{S}(\mathbf{A})$ and $\mathcal{S}(\mathbf{B})$. I.e., for the event that $i \in \mathcal{T}$.

For any $i \in [n]$, let $\tau_{\mathbf{A}}^i$ denote the $k^{\text{th}}$ smallest value of $h(j)/\|\mathbf{A}_j\|_2^2$ over all $j \in [n] \setminus \{i\}$. If $[n] \setminus \{i\}$ has fewer than $k$ values, define $\tau_{\mathbf{a}}^i = \infty$. Define $\tau_{\mathbf{B}}^i$ analogously. The probability that $i \in \mathcal{T} = \mathcal{I}_{\mathbf{A}} \cap \mathcal{I}_{\mathbf{B}}$ conditioned on $\tau^i(\mathbf{A})$ and $\tau^i(\mathbf{B})$ is equal to the probability that *both* $h(i)/\|\mathbf{A}_i\|_2^2 \leq \tau_{\mathbf{A}}^i$ and $h(i)/\|\mathbf{B}_i\|_2^2 \leq \tau_{\mathbf{B}}^i$. I.e., the conditional probability is equal to $\min(1, \|\mathbf{A}_i\|_2^2 \cdot \tau_{\mathbf{A}}^i, \|\mathbf{B}_i\|_2^2 \cdot \tau_{\mathbf{B}}^i)$.

Additionally, observe that, for all sampled $i \in \mathcal{T} = \mathcal{I}_{\mathbf{A}} \cap \mathcal{I}_{\mathbf{B}}$, $\tau_{\mathbf{A}}^i = \tau_{\mathbf{A}}$ and $\tau_{\mathbf{B}}^i = \tau_{\mathbf{B}}$. So, we have:

$$\mathbb{E}\left[\frac{\mathbf{A}_{i,x}\mathbf{B}_{i,y}}{\min(1, \|\mathbf{A}_i\|_2^2 \cdot \tau_{\mathbf{A}}, \|\mathbf{B}_i\|_2^2 \cdot \tau_{\mathbf{B}})}\mathbb{1}_i\right]$$

$$= \mathbb{E}_{\tau_{\mathbf{A}}^i, \tau_{\mathbf{B}}^i}\left[\frac{\mathbf{A}_{i,x}\mathbf{B}_{i,y}}{\min(1, \|\mathbf{A}_i\|_2^2 \cdot \tau_{\mathbf{A}}^i, \|\mathbf{B}_i\|_2^2 \cdot \tau_{\mathbf{B}}^i)} \cdot \Pr\left[i \in \mathcal{T} \mid \tau_{\mathbf{A}}^i, \tau_{\mathbf{B}}^i\right]\right]$$

$$= \mathbf{A}_{i,x}\mathbf{B}_{i,y}.$$

By linearity of expectation, it follows that $\mathbb{E}\left[\mathbf{W}_{x,y}\right] = \sum_{i=1}^{n} \mathbb{E}\left[\frac{\mathbf{A}_{i,x}\mathbf{B}_{i,y}}{\min(1, \|\mathbf{A}_i\|_2^2 \cdot \tau_{\mathbf{A}}^i, \|\mathbf{B}_i\|_2^2 \cdot \tau_{\mathbf{B}}^i)}\mathbb{1}_i\right] = \sum_{i=1}^{n} \mathbf{A}_{i,x}\mathbf{B}_{i,y} = \left[\mathbf{A}^T\mathbf{B}\right]_{x,y}$, as desired.

The next step is to bound the expected squared error. As mentioned earlier, this is made difficult by the fact that $\mathbb{1}_i$ and $\mathbb{1}_j$ are *not* independent random variables. Fortunately, however, we can show that appropriate (random) scalings of these random variables are *uncorrelated*, a technique that is standard in prior analyses of priority sampling for other problems.

In particular, define $p_i = \min(1, \|\mathbf{A}_i\|_2^2 \cdot \tau_{\mathbf{A}}, \|\mathbf{B}_i\|_2^2 \cdot \tau_{\mathbf{B}})$ and define $\tau_{\mathbf{A}}^{i,j}$ to be the $(k-1)^{\text{st}}$ smallest value among $\frac{h(k)}{\|\mathbf{A}_k\|_2}$ for all $k \in [n] \setminus \{i,j\}$. Define $\tau_{\mathbf{B}}^{i,j}$ analogously. We can see that $\Pr[i, j \in \mathcal{T} \mid \tau_{\mathbf{A}}^{i,j}, \tau_{\mathbf{B}}^{i,j}] = \min(1, \|\mathbf{A}_i\|_2^2 \cdot \tau_{\mathbf{A}}^{i,j}, \|\mathbf{B}_i\|_2^2 \cdot \tau_{\mathbf{B}}^{i,j}) \cdot \min(1, \|\mathbf{A}_j\|_2^2 \cdot \tau_{\mathbf{A}}^{i,j}, \|\mathbf{B}_j\|_2^2 \cdot \tau_{\mathbf{B}}^{i,j})$. Moreover, conditioned on $i, j \in \mathcal{T}$, we have that $\tau^{i,j}(\mathbf{A}) = \tau(\mathbf{A})$ and $\tau^{i,j}(\mathbf{B}) = \tau(\mathbf{B})$. So, in particular, conditioned on $i, j \in \mathcal{T}$, $p_i = \min(1, \|\mathbf{A}_i\|_2^2 \cdot \tau_{\mathbf{A}}^{i,j}, \|\mathbf{B}_i\|_2^2 \cdot \tau_{\mathbf{B}}^{i,j})$.

$$\mathbb{E}\left[\frac{\mathbb{1}_i}{p_i}\frac{\mathbb{1}_j}{p_j}\right] = \mathbb{E}_{\tau_{\mathbf{A}}^{i,j}, \tau_{\mathbf{B}}^{i,j}}\left[\frac{1}{\min(1, \|\mathbf{A}_i\|_2^2 \cdot \tau_{\mathbf{A}}^{i,j}, \|\mathbf{B}_i\|_2^2 \cdot \tau_{\mathbf{B}}^{i,j})}\frac{1}{\min(1, \|\mathbf{A}_j\|_2^2 \cdot \tau_{\mathbf{A}}^{i,j}, \|\mathbf{B}_j\|_2^2 \cdot \tau_{\mathbf{B}}^{i,j})}\right.$$

$$\left. \cdot \Pr\left[i, j \in \mathcal{T} \mid \tau_{\mathbf{A}}^{i,j}, \tau_{\mathbf{B}}^{i,j}\right]\right]$$

$$= 1 = \mathbb{E}\left[\frac{\mathbb{1}_i}{p_i}\right]\mathbb{E}\left[\frac{\mathbb{1}_j}{p_j}\right].$$

Since the random variables $\frac{\mathbb{1}_i}{p_i}$ and $\frac{\mathbb{1}_j}{p_j}$ are pairwise uncorrelated for all $i, j$, we have that $\mathbf{A}_{i,x}\mathbf{B}_{i,y} \cdot \frac{\mathbb{1}_i}{p_i}$ and $\mathbf{A}_{j,x}\mathbf{B}_{j,y} \cdot \frac{\mathbb{1}_j}{p_j}$ are pairwise uncorrelated as well. So, we can apply the linearity of variance to conclude:

$$\mathbb{E}[(\mathbf{W}_{x,y} - [\mathbf{A}^T\mathbf{B}]_{x,y})^2] = \mathrm{Var}[\mathbf{W}_{x,y}] = \mathrm{Var}\left[\sum_{i=1}^n \frac{\mathbf{A}_{i,x}\mathbf{B}_{i,y}}{\min(1, \|\mathbf{A}_i\|_2^2 \cdot \tau_\mathbf{A}, \|\mathbf{B}_i\|_2^2 \cdot \tau_\mathbf{B})} \cdot \mathbb{1}_i\right]$$

$$= \sum_{i=1}^n \mathrm{Var}\left[\frac{\mathbf{A}_{i,x}\mathbf{B}_{i,y}}{\min(1, \|\mathbf{A}_i\|_2^2 \cdot \tau_\mathbf{A}, \|\mathbf{B}_i\|_2^2 \cdot \tau_\mathbf{B})} \cdot \mathbb{1}_i\right].$$

So, it suffices to establish individual bounds on $\mathrm{Var}\left[\frac{\mathbf{A}_{i,x}\mathbf{B}_{i,y}}{\min(1, \|\mathbf{A}_i\|_2^2 \cdot \tau_\mathbf{A}, \|\mathbf{B}_i\|_2^2 \cdot \tau_\mathbf{B})} \cdot \mathbb{1}_i\right]$. In order to do so, first observe that, conditioned on $\tau_\mathbf{A}^i, \tau_\mathbf{B}^i$,

$$\mathbb{E}\left[\left(\frac{\mathbf{A}_{i,x}\mathbf{B}_{i,y}}{\min(1, \|\mathbf{A}_i\|_2^2 \cdot \tau_\mathbf{A}, \|\mathbf{B}_i\|_2^2 \cdot \tau_\mathbf{B})} \cdot \mathbb{1}_i\right)^2 \mid \tau_\mathbf{A}^i, \tau_\mathbf{B}^i\right]$$

$$= \left(\frac{\mathbf{A}_{i,x}\mathbf{B}_{i,y}}{\min(1, \|\mathbf{A}_i\|_2^2 \cdot \tau_\mathbf{A}^i, \|\mathbf{B}_i\|_2^2 \cdot \tau_\mathbf{B}^i)}\right)^2 \cdot \min(1, \|\mathbf{A}_i\|_2^2 \cdot \tau_\mathbf{A}^i, \|\mathbf{B}_i\|_2^2 \cdot \tau_\mathbf{B}^i)$$

$$= \frac{\mathbf{A}_{i,x}^2\mathbf{B}_{i,y}^2}{\min(1, \|\mathbf{A}_i\|_2^2 \cdot \tau_\mathbf{A}^i, \|\mathbf{B}_i\|_2^2 \cdot \tau_\mathbf{B}^i)} = \mathbf{A}_{i,x}^2\mathbf{B}_{i,y}^2 \cdot \max\left(1, \frac{1}{\|\mathbf{A}_i\|_2^2 \cdot \tau_\mathbf{A}^i}, \frac{1}{\|\mathbf{B}_i\|_2^2 \cdot \tau_\mathbf{B}^i}\right).$$

We can thus write:

$$\mathrm{Var}\left[\frac{\mathbf{A}_{i,x}\mathbf{B}_{i,y}}{\min(1, \|\mathbf{A}_i\|_2^2 \cdot \tau_\mathbf{A}, \|\mathbf{B}_i\|_2^2 \cdot \tau_\mathbf{B})} \cdot \mathbb{1}_i\right]$$

$$= \mathbf{A}_{i,x}^2\mathbf{B}_{i,y}^2 \cdot \mathbb{E}\left[\max\left(1, \frac{1}{\|\mathbf{A}_i\|_2^2 \cdot \tau_\mathbf{A}^i}, \frac{1}{\|\mathbf{B}_i\|_2^2 \cdot \tau_\mathbf{B}^i}\right)\right] - \mathbf{A}_{i,x}^2\mathbf{B}_{i,y}^2$$

$$= \mathbf{A}_{i,x}^2\mathbf{B}_{i,y}^2 \cdot \mathbb{E}\left[\max\left(0, \frac{1}{\|\mathbf{A}_i\|_2^2 \cdot \tau_\mathbf{A}^i} - 1, \frac{1}{\|\mathbf{B}_i\|_2^2 \cdot \tau_\mathbf{B}^i} - 1\right)\right]$$

$$\leq \mathbf{A}_{i,x}^2\mathbf{B}_{i,y}^2 \cdot \mathbb{E}\left[\max\left(\frac{1}{\|\mathbf{A}_i\|_2^2 \cdot \tau_\mathbf{A}^i}, \frac{1}{\|\mathbf{B}_i\|_2^2 \cdot \tau_\mathbf{B}^i}\right)\right]$$

$$\leq \mathbf{A}_{i,x}^2\mathbf{B}_{i,y}^2 \cdot \mathbb{E}\left[\frac{1}{\|\mathbf{A}_i\|_2^2 \cdot \tau_\mathbf{A}^i}\right] + \mathbf{A}_{i,x}^2\mathbf{B}_{i,y}^2 \cdot \mathbb{E}\left[\frac{1}{\|\mathbf{B}_i\|_2^2 \cdot \tau_\mathbf{B}^i}\right]$$

$$= \frac{\mathbf{A}_{i,x}^2\mathbf{B}_{i,y}^2}{\|\mathbf{A}_i\|_2^2} \cdot \mathbb{E}\left[\frac{1}{\tau_\mathbf{A}^i}\right] + \frac{\mathbf{A}_{i,x}^2\mathbf{B}_{i,y}^2}{\|\mathbf{B}_i\|_2^2} \cdot \mathbb{E}\left[\frac{1}{\tau_\mathbf{B}^i}\right].$$

We can apply Claim 5 from Daliri et al. (2024a) to bound $\mathbb{E}\left[\frac{1}{\tau_\mathbf{A}^i}\right] \leq \frac{\|\mathbf{A}\|_F^2}{k-1}$ and $\mathbb{E}\left[\frac{1}{\tau_\mathbf{B}^i}\right] \leq \frac{\|\mathbf{B}\|_F^2}{k-1}$. So we have:

$$\mathbb{E}[(\mathbf{W}_{x,y} - [\mathbf{A}^T\mathbf{B}]_{x,y})^2] \leq \sum_{i=1}^n \frac{\mathbf{A}_{i,x}^2\mathbf{B}_{i,y}^2}{\|\mathbf{A}_i\|_2^2} \cdot \mathbb{E}\left[\frac{1}{\tau_\mathbf{A}^i}\right] + \frac{\mathbf{A}_{i,x}^2\mathbf{B}_{i,y}^2}{\|\mathbf{B}_i\|_2^2} \cdot \mathbb{E}\left[\frac{1}{\tau_\mathbf{B}^i}\right)\right]$$

$$\leq \sum_{i=1}^n \frac{\mathbf{A}_{i,x}^2\mathbf{B}_{i,y}^2}{\|\mathbf{A}_i\|_2^2} \cdot \frac{\|\mathbf{A}\|_F^2}{k-1} + \frac{\mathbf{A}_{i,x}^2\mathbf{B}_{i,y}^2}{\|\mathbf{B}_i\|_2^2} \cdot \frac{\|\mathbf{B}\|_F^2}{k-1}$$

$$= \sum_{i=1}^n \frac{\mathbf{A}_{i,x}^2\mathbf{B}_{i,y}^2}{k-1} \cdot \left(\frac{\|\mathbf{A}\|_F^2}{\|\mathbf{A}_i\|_2^2} + \frac{\|\mathbf{B}\|_F^2}{\|\mathbf{B}_i\|_2^2}\right),$$

as desired. □

## B  MATRIX PRODUCT SKETCHING WITH THRESHOLD SAMPLING

We motivated our Priority Sampling method from Section 2 via a simpler matrix sketching method based on *Threshold Sampling*. We include a full analysis of this method here for pedagogical purposes,

since the method is much easier to analysis. However, in general, we believe that Priority Sampling is preferable since it offers a fixed size sketch. In terms of accuracy, recent work on inner product sketching finds that both methods perform nearly identically to each other (Daliri et al., 2024b). Experiments suggest the same is true for general matrix-matrix product sketching.

**Sketching.** As discussed, Threshold Sampling uses a shared hash function $h : [n] \to [0, 1]$, which is assumed to be uniformly random. As shown in the pseudocode in Algorithm 4, the method selects all rows from $\mathbf{A}$ for which $h(i)/\|\mathbf{A}_i\|_2^2$ falls below a fixed "global threshold", $\tau_{\mathbf{A}} = k/\|\mathbf{A}\|_F^2$. Here, $k$ is a parameter that determines the size of the sketch $\mathcal{S}(\mathbf{A})$ produced by Algorithm 4. There will be at most $k$ rows selected in expectation, but the exact number depends on the random choice of $h$.

---

**Algorithm 4** Threshold Sampling

---

**Input:** Matrix $\mathbf{A}$ of size $n \times d$, random seed $s$, target number of row samples, $k$.
**Output:** Sketch $\mathcal{S}(\mathbf{A}) = \{\mathcal{I}_{\mathbf{A}}, V_{\mathbf{A}}, \tau_{\mathbf{A}}\}$, where $\mathcal{I}_{\mathbf{A}}$ is a subset of row indices from $\{1, \ldots, n\}$ and
$\quad$ $V_{\mathbf{A}}$ contains $\mathbf{A}_i$ for all $i \in \mathcal{I}_{\mathbf{A}}$.

---

1: Use random seed $s$ to select a uniformly random hash function $h : \{1, ..., n\} \to [0, 1]$.
2: Initialize $\mathcal{I}_{\mathbf{A}}$ and $V_{\mathbf{A}}$ to be empty lists.
3: **for** $i \in 1, \ldots, n$ **do**
4: $\quad$ Set threshold $\tau_i = k \cdot \frac{\|\mathbf{A}_i\|_2^2}{\|\mathbf{A}\|_F^2}$.
5: $\quad$ **if** $h(i) \leq \tau_i$ **then**
6: $\quad\quad$ Append $i$ to $\mathcal{I}_{\mathbf{A}}$, append $\mathbf{A}_i$ to $V_{\mathbf{A}}$.
7: **return** $\mathcal{S}(\mathbf{A}) = \{\mathcal{I}_{\mathbf{A}}, V_{\mathbf{A}}, \tau_{\mathbf{A}}\}$ where $\tau_{\mathbf{A}} = k/\|\mathbf{A}\|_F^2$.

---

**Estimation.** Similar to Priority Sampling, after constructing our sketches $\mathcal{S}(\mathbf{A})$ and $\mathcal{S}(\mathbf{B})$, we approximate the matrix product of $\mathbf{A}$ and $\mathbf{B}$ by computing a weighted sum of outerproducts of rows included in both $\mathcal{S}(\mathbf{A})$ and $\mathcal{S}(\mathbf{B})$. In fact, we can use the exact same procedure defined in Algorithm 2 from Section 2.

Unlike Priority Sampling, Threshold Sampling ensures that the probability of sampling any given row $\mathbf{A}_i$ is an independent random event. There is no dependence on the event that another row $j$ gets sampled. In particular, we can easily compute the probability of index $i$ being included in both sketches $\mathcal{S}(\mathbf{A})$ and $\mathcal{S}(\mathbf{B})$. It is exactly equal to $p_i = \min\left(1, k \cdot \frac{\|\mathbf{A}_i\|_2^2}{\|\mathbf{A}\|_F^2}, k \cdot \frac{\|\mathbf{B}_i\|_2^2}{\|\mathbf{B}\|_F^2}\right)$.

**Guarantees.** Our primary theoretical guarantee for Threshold Sampling can be stated as follows:

**Theorem 6.** *Let $\mathbf{A} \in \mathbb{R}^{n \times d}$, $\mathbf{B} \in \mathbb{R}^{n \times m}$, and let $\mathcal{S}(\mathbf{A}) = \{\mathcal{I}_{\mathbf{A}}, V_{\mathbf{A}}, \tau_{\mathbf{A}}\}$ and $\mathcal{S}(\mathbf{B}) = \{\mathcal{I}_{\mathbf{B}}, V_{\mathbf{B}}, \tau_{\mathbf{B}}\}$ be sketches produced by Algorithm 4 with input $k$ and a shared seed $s$. Suppose $\mathbf{W}$ is the approximate matrix of $\mathbf{A}^T\mathbf{B}$ calculated using Algorithm 2 on these sketches. Then, $\mathbb{E}[\mathbf{W}] = \mathbf{A}^T\mathbf{B}$ and*

$$\mathbb{E}\left[\|\mathbf{W} - \mathbf{A}^T\mathbf{B}\|_F^2\right] \leq \frac{2}{k}\|\mathbf{A}\|_F^2\|\mathbf{B}\|_F^2.$$

*Additionally, $\mathbb{E}[|\mathcal{I}_{\mathbf{A}}|] \leq k$ and $\mathbb{E}[|\mathcal{I}_{\mathbf{B}}|] \leq k$. I.e., each sketch contains no more than $k$ row indices in expectation.*

Theorem 6 essentially matches Theorem 4, although is actually a bit tighter, as the $2/(k-1)$ prefactor is replaced with $2/k$. The only disadvantage of the theorem is that we do not have a fixed upper bound on $|\mathcal{I}_{\mathbf{A}}|$ and $|\mathcal{I}_{\mathbf{B}}|$, which are equal to the number of rows sampled from $\mathbf{A}$ and $\mathbf{B}$, respectively. We also remark that, as for Theorem 4, Theorem 6 an be combined with Markov's inequality to give a high probability bound: if we set $k = \frac{2/\delta}{\epsilon^2}$ then we achieve error $\|\mathbf{W} - \mathbf{A}^T\mathbf{B}\|_F \leq \epsilon\|\mathbf{A}\|_F\|\mathbf{B}\|_F$ with probability $1 - \delta$.

*Proof of Theorem 6.* Let $\mathbb{1}_i$ denote the indicator random variable for the event that $i$ is included in *both* $\mathcal{I}_{\mathbf{A}}$ and $\mathcal{I}_{\mathbf{B}}$. $\mathbb{1}_i = 1$ if this event occurs and 0 if it does not. Note that, for $i \neq j$, $\mathbb{1}_i$ is independent from $\mathbb{1}_j$, since the hash values $h(i)$ and $h(j)$ are drawn uniformly and independently from $[0, 1]$. Moreover, we claim that $\mathbb{1}_i$ is equal to 1 with probability:

$$p_i = \min\left(1, \frac{k \cdot \|\mathbf{A}_i\|_2^2}{\|\mathbf{A}\|_F^2}, \frac{k \cdot \|\mathbf{B}_i\|_2^2}{\|\mathbf{B}\|_F^2}\right) = \min(1, \tau_{\mathbf{A}} \cdot \|\mathbf{A}_i\|_2^2, \tau_{\mathbf{B}} \cdot \|\mathbf{B}_i\|_2^2). \tag{2}$$

This is because for index $i$ to be included in both $\mathcal{I}_\mathbf{A}$ and $\mathcal{I}_\mathbf{B}$, $h(i)$ must be less than both $\|\mathbf{A}_i\|_2^2/\|\mathbf{A}\|_F^2$ and $\|\mathbf{B}_i\|_2^2/\|\mathbf{B}\|_F^2$ simultaneously (see line 3 of Algorithm 4). Given the probability of sampling each item, we can find the expectation of the approximation $\mathbf{W}$.

$$\mathbb{E}[\mathbf{W}] = \sum_{i=1}^n p_i \cdot \frac{\mathbf{A}_i \mathbf{B}_i^T}{p_i} = \sum_{i=1}^n \mathbf{A}_i \mathbf{B}_i^T = \mathbf{A}^T \mathbf{B}. \tag{3}$$

This proves the desired claim on the expection of $\mathbf{W}$. It is left to bound the $\mathbb{E}\left[\|\mathbf{W} - \mathbf{A}^T \mathbf{B}\|_F^2\right]$.

We do so by bounding the squared error of each entry in $\mathbf{W}$ separately. In particular, for the $x, y$ entry, we write:

$$\mathbb{E}\left[\left(\mathbf{W}_{x,y} - [\mathbf{A}^T\mathbf{B}]_{x,y}\right)^2\right] = \mathrm{Var}\left[\mathbf{W}_{x,y}\right] = \mathrm{Var}\left[\sum_{i=1}^n \mathbb{1}_i \frac{\mathbf{A}_{i,x}\mathbf{B}_{i,y}}{p_i}\right] = \sum_{i=1}^n \mathrm{Var}\left[\mathbb{1}_i \frac{\mathbf{A}_{i,x}\mathbf{B}_{i,y}}{p_i}\right].$$

Above we use the fact that $\mathbb{1}_1, \dots, \mathbb{1}_n$ are independent to apply linearity of variance. Let $\mathcal{H}$ denote the the set of all $i$ for which $p_i \neq 0$ and $p_i \neq 1$. Using that $\mathrm{Var}[\mathbb{1}_i] = p_i(1 - p_i)$, we have:

$$\mathbb{E}\left[\left(\mathbf{W}_{x,y} - [\mathbf{A}^T\mathbf{B}]_{x,y}\right)^2\right] = \sum_{i\in\mathcal{H}} \frac{\mathbf{A}_{i,x}^2\mathbf{B}_{i,y}^2}{p_i^2} \cdot p_i(1 - p_i) \leq \sum_{i\in\mathcal{H}} \frac{\mathbf{A}_{i,x}^2\mathbf{B}_{i,y}^2}{p_i}.$$

So we can bound $\mathbb{E}\left[\|\mathbf{W} - \mathbf{A}^T\mathbf{B}\|_F^2\right]$ by:

$$\mathbb{E}\left[\|\mathbf{W} - \mathbf{A}^T\mathbf{B}\|_F^2\right] = \sum_{x=1}^d \sum_{y=1}^m \mathbb{E}\left[\left(\mathbf{W}_{x,y} - [\mathbf{A}^T\mathbf{B}]_{x,y}\right)^2\right]$$

$$\leq \sum_{x=1}^d \sum_{y=1}^m \sum_{i\in\mathcal{H}} \frac{\mathbf{A}_{i,x}^2 \cdot \mathbf{B}_{i,y}^2}{p_i} = \sum_{i\in\mathcal{H}} \frac{1}{p_i} \sum_{x=1}^d \mathbf{A}_{i,x}^2 \sum_{y=1}^m \mathbf{B}_{i,y}^2$$

$$= \sum_{i\in\mathcal{H}} \frac{1}{p_i}\|\mathbf{A}_i\|_2^2\|\mathbf{B}_i\|_2^2 = \sum_{i\in\mathcal{H}} \frac{\|\mathbf{B}_i\|_2^2 \cdot \|\mathbf{A}_i\|_2^2}{\min\left(\frac{k\cdot\|\mathbf{A}_i\|_2^2}{\|\mathbf{A}\|_F^2}, \frac{k\cdot\|\mathbf{B}_i\|_2^2}{\|\mathbf{B}\|_F^2}\right)}.$$

Recall that we defined $p_i = \min\left(1, \frac{k\cdot\|\mathbf{A}_i\|_2^2}{\|\mathbf{A}\|_F^2}, \frac{k\cdot\|\mathbf{B}_i\|_2^2}{\|\mathbf{B}\|_F^2}\right)$, so in the last step, we have used the fact that $p_i \neq 1$ for $i \in \mathcal{H}$. Continuing, we can bound:

$$\mathbb{E}\left[\|\mathbf{W} - \mathbf{A}^T\mathbf{B}\|_F^2\right] \leq \sum_{i\in\mathcal{H}} \|\mathbf{A}\|_F^2\|\mathbf{B}\|_F^2 \frac{\frac{\|\mathbf{B}_i\|_2^2}{\|\mathbf{B}\|_F^2} \cdot \frac{\|\mathbf{A}_i\|_2^2}{\|\mathbf{A}\|_F^2}}{\min\left(\frac{k\cdot\|\mathbf{A}_i\|_2^2}{\|\mathbf{A}\|_F^2}, \frac{k\cdot\|\mathbf{B}_i\|_2^2}{\|\mathbf{B}\|_F^2}\right)}$$

$$\leq \sum_{i\in\mathcal{H}} \|\mathbf{A}\|_F^2\|\mathbf{B}\|_F^2 \frac{\max(\|\mathbf{A}_i\|_2^2/\|\mathbf{A}\|_F^2, \|\mathbf{B}_i\|_2^2/\|\mathbf{B}\|_F^2)}{k}$$

$$\leq \frac{\|\mathbf{A}\|_F^2\|\mathbf{B}\|_F^2}{k} \sum_{i\in\mathcal{H}}\frac{\|\mathbf{A}_i\|_2^2}{\|\mathbf{A}\|_F^2} + \frac{\|\mathbf{B}_i\|_2^2}{\|\mathbf{B}\|_F^2} \leq \frac{2}{k}\|\mathbf{B}\|_F^2\|\mathbf{A}\|_F^2.$$

In the second to last step, we upper bounded the maximum but the sum. $\qquad\square$

## C  FURTHER EXPERIMENTS ON ATTENTION MODELS

One key reason for not sketching the Query matrix in this application is that it is not quantized, unlike the Key and Value matrices. This means there is no need to apply sketching techniques to the Query matrix, as it is recalculated for each input token and does not benefit from compression methods used on more static, larger matrices. An important advantage of using the sampling method is that it allows selective sketching of matrices that are both large and have a static or slow-changing nature, such as the Key matrix. In contrast, linear sketching requires projecting both the Query and Key matrices, regardless of their size or dynamism. In our experiments, we had access to the entire Queries matrix while we applied Priority Sampling to sketch the considerably larger Key matrix. This approach effectively demonstrates the efficiency of sampling in handling large-scale data while preserving the dynamic properties of the Query matrix in real-time applications.

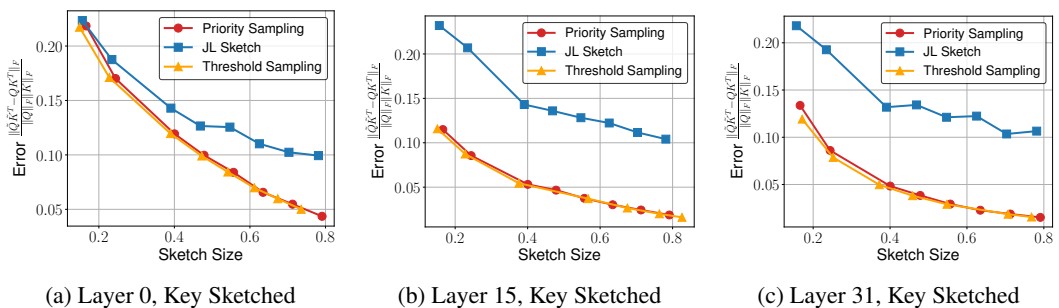

(a) Layer 0, Key Sketched     (b) Layer 15, Key Sketched     (c) Layer 31, Key Sketched

Figure 5: Comparison of KV Cache Sketching Methods on the LongBench for `MultiFieldQA`: The plots illustrate the accuracy of various sketching methods in approximating $\mathbf{Q}\mathbf{K}^T$ across different sketch sizes. The Query matrix remains untouched, and only the Key matrices $\mathbf{K}$ are sketched using Priority Sampling and Threshold Sampling, whereas the JL sketch requires the projection of both matrices $\mathbf{Q}, \mathbf{K}$. Layers refer to the individual layers (hidden layers) of the Transformer architecture in the LLaMA 2 model.

