# OpenReview forum: "Matrix Product Sketching via Coordinated Sampling"
_ICLR.cc/2025/Conference — ICLR 2025 Poster_

### Official Review · Reviewer_Ah4Z · 2024-10-17

**Soundness:** 3
**Presentation:** 3
**Contribution:** 2
**Rating:** 5
**Confidence:** 4

**Summary:**

Given two matrices $\mathbf{A} \in \mathbb{R}^{n \times d}$ and $\mathbf{B} \in \mathbb{R}^{n \times m}$, the problem of approximately computing the product $\mathbf{A}^\top \mathbf{B}$ using sketching techniques has been extensively studied, with a wide range of applications in machine learning and numerical linear algebra. The existing work includes two types of sketching methods: (1) Generating a random matrix $\boldsymbol{\Pi} \in \mathbb{R}^{k \times n}$ such that $(\boldsymbol \Pi \mathbf{A})^\top (\boldsymbol \Pi \mathbf{B}) \approx \mathbf{A}^\top \mathbf{B}$; (2) Sampling and reweighting the rows of $\mathbf{A}$ and $\mathbf{B}$ to obtain $\widetilde{\mathbf{A}}$ and $\widetilde{\mathbf{B}}$ such that $\widetilde{\mathbf{A}}^\top \widetilde{\mathbf{B}} \approx \mathbf{A}^\top \mathbf{B}$. This paper studies matrix multiplication in the $\textit{sketching setting}$, which requires the sketches $\mathcal{S}(\mathbf{A})$ and $\mathcal{S}(\mathbf{B})$ of $\mathbf{A}$ and $\mathbf{B}$ are computed independently. Nevertheless, the classical sampling method does not work as the rows are sampled with non-uniform probabilities. This paper applies the sampling method effectively in the sketching setting by virtue of the $\textit{coordinated sampling method}$ (aka $\textit{priority sampling}$), which ensures that the number of commonly sampled row indices from $\mathbf{A}$ and $\mathbf{B}$ is guaranteed. Furthermore, the proposed algorithm achieves the same guarantee as the existing sampling method and is more efficient when the two input matrices are sparse. In addition, the developed method can be extended to $\ell_2$ regression. The experimental results prove the effectiveness of the proposed algorithm on synthetic and real datasets.

**Strengths:**

(1) This paper studies the approximate matrix multiplication in the sketching setting by the coordinated sampling technique. The proposed algorithm matches the guarantee of the existing sampling-based method and is more efficient when the input matrices are sparse.
(2) The developed algorithm also works for $\ell_2$ regression problem and improves the existing bound when the input matrix is sparse.
(3) This paper is well-written and easy to understand.

**Weaknesses:**

(1) In essence, the coordinated sampling technique is nearly the same as the threshold sampling method and has also been applied in other sketching work [DLT'07, DFMSZ'24]. From the technique perspective, this paper's contribution is kind of limited.
(2) The restriction in sketching setting often occurs when communications matter, like in distributed systems. Although this paper proposes an algorithm that works in the sketching setting, there are no specific applications in distributed sketching.
(3) This paper claims that priority sampling is more complicated than threshold sampling whose sketch size can only be bounded in expectation. However, in Figure 1 and Figure 4, we can see that priority sampling does not outperform threshold sampling.

**Questions:**

In Algorithm 3, it requires computing the leverage score of each row of $\mathbf{A}$ (i.e., $\mathbf{A}_i (\mathbf{A}^\top \mathbf{A})^{-1} \mathbf{A}_i^\top$) and $(\mathbf{A}^\top \mathbf{A})^{-1}$.
(1) What's their running time?
(2) Can you compare this algorithm with the linear regression in the work of Clarkson and Woodruff [CW'13]?
(3) For sketched regression, do we only care about the size of sketches? If so, in Section 1.4, it needs to compute $\min {\mathbf{x}} ||\mathbf{A}^i \mathbf{x} - \mathbf{b}||_2$ for each $i$. Then the running time could be more crucial.

---

> ### Author Response · Authors · 2024-11-19
> **Response to Reviewer Ah4Z's Comments**
>
> We appreciate your thoughtful suggestions and detailed comments to help enhance the clarity of the paper.
>
> ---
>
> ***
>
> **Weakness**:
>
> 1- We address this question in our response to `Reviewer NkFE` as well. That response is pasted below for convenience.
>
> *Regarding Priority Sampling, the main difference between the prior analysis for inner products is as follows: The matrix-matrix product $A^TB$ can be viewed as a set of $d\times m$ inner products between the $d$ columns of $A$ and $m$ columns of $B$. Consider two fixed columns, $a$ from $A$ and $b$ from $B$. In contrast to the prior work, we do not sample the $i^\text{th}$ entries $a_i$ and $b_i$ with probability proportion to their squared magnitude – the sampling probability depends on the norm of the entire $i^\text{th}$ row in $A$ and $B$. As a result, we cannot prove our result by proving a “per inner product” guarantee – for some column pairs, the method will undersample heavy entries, and thus have high error. Instead, we must bound the total variance across all $d\times m$ inner products. Ultimately, doing so is not too difficult: the analysis of priority sampling is still < 2 pages, and all “from first principles”. However, we hope the simplicity of the methods and analysis can be an advantage of the result.*
>
> ---
>
> 2- We have also addressed this question in our response to `Reviewer ev8D`. For your convenience, we have included that response below.
>
> *The issue with communication arises when we wish to compress multiple different matrices and to compute approximate matrix-matrix products between multiple pairs of sketches. For example, suppose we have a database of matrices $A_1,\ldots, A_q$ that need to be “pre-compressed” (as is the case in dataset search or multi-vector retrieval). At some later time, we receive a “query matrix” $B$, and wish to compute $A_1^TB, \ldots, A_q^TB$ based on a sketch of $B$. Because $A_1, \ldots, A_q$ could have very different row norms, a norm based sampling method will not sample the same set of rows from the matrices. So it is not clear what set of indices to communicate that will allow us to create a single sketch of $B$ that can be used when approximating each product $A_1^TB, \ldots, A_q^TB$. In fact, given how different $A_1, \ldots, A_q$ could be, our intuition is that, to achieve worst case guarantees, the sketch of $B$ should not depend on any individual $A_i$, necessitating communication-free sketching. We will think about if this claim can be made formal, but the main point is that naive “index sharing” approaches do not work in such settings. There might be alternative methods that use communication in a more careful way, but this would be surprising to us.*
>
> We will work on clarifying this further in the introduction, as these settings are partially addressed in Section 1.4.
>
> ---
>
> 3-  The reviewer is correct that, in terms of (expected) sketch size vs. accuracy, Priority Sampling is not better than Threshold Sampling. However, it has the advantage of a fixed sketch size, which is very important in applications, where a fixed amount of memory would typically be required for each sketch. For example, in vector-search applications, sketch sizes are often carefully chosen due to memory considerations (e.g., a certain number of sketched vectors should fit in L1 or L2 cache). We expect the same sort of requirements to arise when sketching matrices, which is why we analyze priority sampling.
>
> ---
>
> ***
>
> **Questions**:
>
> 1- Naively, computing $(A^T A)^{-1}$ takes $O(n \cdot d^2 + d^3)$, and calculating the leverage scores for all rows takes $O(n \cdot d^2)$ , resulting in a total running time of $O(d^3 + n \cdot d^2)$. This runtime can be accelerated with more advanced techniques (e.g., the Clarkson-Woodruff 2013 paper mentioned by the reviewer), ultimately resulting in the $nd^2$ term being replaced with with $nnz(A)\log(n)$, where $nnz(A)$ is the number of non-zero entries in $A$. We will add some discussion about this to the paper.
>
> ---
>
> 2- This is a good question. The regression algorithm from Clarkson-Woodruff 2013 is a “linear sketching” method. As discussed below Theorem 3, it would lead to a sketch size of $O(d^2/\epsilon)$ for the regression problem, whereas we achieve size $O(sd/\epsilon)$. [CW13] does have an advantage over other linear sketching methods in that the sketching matrix, $S$, can be applied extremely quickly (roughly in $O(nnz(A))$ time). However, the sketch size is no better (and in fact, slightly worse) than if a slower random Gaussian or random sign sketch was used instead. We are primarily focused on sketched size for this paper, as in the settings we consider such as multi-vector retrieval, sketch time is less important (it is a preprocessing step).
>
> ---

---

> ### Author Response · Authors · 2024-11-19
> **Response to Reviewer Ah4Z's Comments (Continued)**
>
> 3- As mentioned, for this paper, we are primarily focused on sketch size: in practical applications, there is often a strong connection between reducing memory requirements and improving wall-clock runtime. By working with smaller sampled matrices $S(A_i)$ and $S(b)$, instead of the full matrices $A_i$ and $b$, the computational cost of solving $\min_x || S(A_i)^T x - S(b) ||_2$ is significantly reduced. This reduction in matrix size directly translates to faster computations, making sketching not just memory-efficient but also runtime-efficient. That said, in future work, it might be interesting to consider if there are even faster estimation procedures that avoid computing $A_i^+$ for each $A_i$.

---

> > ### Comment · Reviewer_Ah4Z · 2024-11-22
> >
> > Thank the authors for the detailed feedback! I read all the reviewers' comments and the authors' feedback. In summary, this paper proposes algorithms applied for sketching setting, which is interesting and practical. From the perspective of technique, the novelty of priority sampling compared with existing methods is kind of limited. Based on this, I will retain my score.

---

### Official Review · Reviewer_ev8D · 2024-11-04

**Soundness:** 3
**Presentation:** 3
**Contribution:** 2
**Rating:** 6
**Confidence:** 4

**Summary:**

The paper revisits the approximation matrix product problem. That is, given two matrix $A, B$, we want to compute a matrix $W$ in a fast time where $\|W - A^T B\|_F^2 \le \|A\|_F^2 \|B\|_F^2$. The paper studies the case where sketches must be computed independently of each other, except for the use of a shared random seed. The paper provides a new method that is based on coordinated random sampling and has sketch size $O(s/\epsilon^2)$ assuming $A$ and $B$ have $s$ non-zero entries per row. Based on this, the paper next proposes a new algorithm for the $\ell_2$ regression problem. Finally, the paper gives an empirical evaluation that demonstrates the advantage of the proposed algorithm.

**Strengths:**

- The theoretical analysis of this paper is solid. The paper gives a new sketching algorithm with size $O(s^2 /\epsilon^2)$. This bound will be better for sparse matrix compared to the previous methods, which is interesting to me.

- The paper gives a detailed experiment that demonstrates the advantage of the proposed algorithms.

- The presentation of the paper is good. The paper has a nice introduction section.

**Weaknesses:**

- I still do not understand the motivation of the new model the paper discusses well (see the questions below). Maybe the authors can give more explanation about this？

- It will be better if the experiments can also give a comparison to the previous sampling-based method.

**Questions:**

Approximation Matrix Product: I am not sure I fully understand the motivation of the model considered, as mentioned by the authors, sampling to either $\|A_i\|_2$ and $\|B_i\|_2 $ can also get a good bound. The communication about the probability can be somewhat expensive but we can also sample from $A$ and then tell the other part the indices of the sampled rows, in which case we will have much better communication cost?

Regression: there are some recent works that study the active regression problem where the goal is to access as fewer number of the entries of $b$ as possible, see, e.g, [1, 2]. In particular, it has been shown that sample $O(d/\epsilon)$ rows of $A$(where the sample probability is only computed from the matrix) can give a $(1 + \epsilon)$ relative error of the solutions. Can the authors give a comparison of both models?

Minor:

1.Line 238: that is $k \cdot |B_i|^2 / |B|_F^2$ ?

[1] Cameron Musco et al. Active Linear Regression for $\ell_p$ Norms and Beyond.

[2] Xue Chen et al. Active regression via linear-sample sparsification active regression via linear-sample sparsification.

---

> ### Author Response · Authors · 2024-11-19
> **Response to Reviewer ev8D's Comments**
>
> We appreciate the reviewer's constructive feedback and respond to specific questions below:
>
> ---
>
> ***
>
> **Weakness**:
>
> Regarding comparing to “the previous sampling-based method”, we are not quite sure what method the reviewer refers to. The previous methods based on weighted sampling cannot be implemented in the “sketching model” where communication is not allowed. Uniform row sampling can be implemented in the sketching model – we will consider adding an experiment comparing our approach to uniform sampling, although our experience is that uniform sampling often performs far worse than the norm-based sampling methods or JL.
>
> ---
>
> ***
>
> **Questions**:
>
> 1- The reviewer is correct that, in some settings, communication is not a major concern – we simply need to communicate the indices of the rows sampled. However, in other settings, communication is clearly not useful, which is why we seek algorithms that do not require any. We will try to clarify this further in the introduction. Such settings are already discussed in Section 1.4 to an extent.
>
>
> The issue with communication arises when we wish to compress multiple different matrices and to compute approximate matrix-matrix products between multiple pairs of sketches. For example, suppose we have a database of matrices $A_1,\ldots, A_q$ that need to be “pre-compressed” (as is the case in dataset search or multi-vector retrieval). At some later time, we receive a “query matrix” $B$, and wish to compute $A_1^TB, \ldots, A_q^TB$ based on a sketch of $B$. Because $A_1, \ldots, A_q$ could have very different row norms, a norm based sampling method will not sample the same set of rows from the matrices. So it is not clear what set of indices to communicate that will allow us to create a single sketch of $B$ that can be used when approximating each product $A_1^TB, \ldots, A_q^TB$. In fact, given how different $A_1, \ldots, A_q$ could be, our intuition is that, to achieve worst case guarantees, the sketch of $B$ should not depend on any individual $A_i$, necessitating communication-free sketching. We will think about if this claim can be made formal, but the main point is that naive “index sharing” approaches do not work in such settings. There might be alternative methods that use communication in a more careful way, but this would be surprising to us.
>
> ---
>
> 2- This is a good question. In the active learning setting, the goal is also to sample rows from $A$ and corresponding entries from $b$, although under different restrictions than our paper. In particular, in the active learning setting, $b$ is unknown, so the values of its entries cannot be taken into account when choosing which indices to select. However, the indices selected from $b$ can depend on which rows are selected from $A$ (in fact, the methods cited by the reviewer sample rows from $A$ using a probability distribution that depends only from $A$, and then select the corresponding entries from $b$ deterministically).
>
>
> In contrast, in our setting, $b$ is known so the value of its entries can be taken into account when selecting which entries to subsample. However, we do not allow the sample taken from $b$ to depend on those taken from $A$, and vice versa. It seems that our setting is more challenging, in that it is not possible to obtain a relative error guarantee for regression using $o(n)$ samples, let alone $O(d/\epsilon)$ samples as in the active regression setting. **This point is discussed further in our response to Reviewer St9U.** Nevertheless, at least we are able to obtain some non-trivial additive approximation guarantee (Theorem 3).

---

> > ### Comment · Reviewer_ev8D · 2024-11-27
> >
> > Thanks for the detailed response. I will keep my score $6$.
> >
> > > The previous methods based on weighted sampling cannot be implemented in the “sketching model” where communication is not allowed.
> >
> > As pointed out by the authors, the proposed algorithm has a better bound for the sparse matrix. Hence, in my opinion it is still interesting to compare with the previous sampling-based algorithms in the normal setting.

---

### Official Review · Reviewer_NkFE · 2024-11-06

**Soundness:** 3
**Presentation:** 4
**Contribution:** 3
**Rating:** 6
**Confidence:** 3

**Summary:**

Overview: The paper presents a method for sketching two matrices A and B, each held by a different party, such that the product of the matrices can be approximated up to an additive term of eps ||A||||B|| from the sketches alone. Each sketch consists of k rows/columns of the sketched matrix, for k=O(1/eps^2). For sparse matrices, such sketches can be more compact than the state-of-the-art linear sketches obtained via random projections.

Techniques:  the paper presents two sketching methods: Threshold Sampling and Priority Sampling. Both methods are variants of coordinated sampling, where a shared random seed is used to sample columns/rows in a correlated manner. Threshold Sampling is simpler to analyze, while Priority Sampling offers better control over the sketch size.  In addition, the paper presents two applications of the proposed sketches, to multi-vector retrieval and regression-based dataset search.

The authors complement the theoretical development with empirical evaluation of their algorithms. In particular, both sketching methods are shown to improve (for sparse matrices) over the standard random projection approach, as predicted theoretically.

**Strengths:**

S1: Interesting problem.
S2: Elegant solutions.
S3: Solid experiments.

**Weaknesses:**

W1: The result and the approach are not very surprising, given the prior work of Bessa et al and Daliri et al
W2: The analysis of one of the algorithms (Threshold Sampling) seems fairly straightforward.

**Questions:**

Q: In the paper you write "In doing so, we encounter several technical challenges, including the fact that Priority Sampling—being a without-replacement sampling procedure—generates non-i.i.d. row samples from A and B.". Can you list some of the other technical challenges as well ?

---

> ### Author Response · Authors · 2024-11-19
> **Response to Reviewer NkFE's Comments**
>
> We thank the reviewer for their thoughtful feedback. Indeed, the Threshold Sampling analysis is straightforward, although the method has a disadvantage in that it does not guarantee a sketch of a fixed size (the sketch size is random, albeit bounded in expectation).
>
>
> Regarding Priority Sampling, the main difference between the prior analysis for inner products is as follows: The matrix-matrix product $A^TB$ can be viewed as a set of $d\times m$ inner products between the $d$ columns of $A$ and $m$ columns of $B$. Consider two fixed columns, $a$ from $A$ and $b$ from $B$. In contrast to the prior work, we do not sample the $i^\text{th}$ entries $a_i$ and $b_i$ with probability proportion to their squared magnitude – the sampling probability depends on the norm of the entire $i^\text{th}$ row in $A$ and $B$. As a result, we cannot prove our result by proving a “per inner product” guarantee – for some column pairs, the method will undersample heavy entries, and thus have high error. Instead, we must bound the total variance across all $d\times m$ inner products. Ultimately, doing so is not too difficult: the analysis of priority sampling is still < 2 pages, and all “from first principles”. However, we hope the simplicity of the methods and analysis can be an advantage of the result.

---

> > ### Comment · Reviewer_NkFE · 2024-11-26
> > **Thank you for the rebuttal**
> >
> > Thank you for elaborating on the technical challenges. Indeed, I view the simplicity of the algorithms and analysis to be a plus. I support acceptance of this paper.

---

### Official Review · Reviewer_St9U · 2024-11-11

**Soundness:** 4
**Presentation:** 4
**Contribution:** 3
**Rating:** 6
**Confidence:** 4

**Summary:**

This paper addresses the problem of approximating the matrix product $A^T B$ using sketching-based techniques in a new setting with two key requirements: (1) preserving the potential sparsity of $A$ and $B$ and (2) ensuring that sketches of $A$ and $B$ can be computed independently—motivated by scenarios like distributed computing and multi-vector retrieval. Traditional linear sketches, often relying on dense random matrices, fail to preserve sparsity. However, sparsity preservation (requirement 1) can be achieved using importance sampling techniques, where rows of $A$ and $B$ are subsampled and reweighted. Unfortunately, existing importance sampling methods require knowledge of both $A$ and $B$ simultaneously, which contradicts requirement (2).

To address this, the paper proposes a new approach called coordinated random sampling, which only requires shared randomness to independently compute sketches of $A$ and $B$ by row subsampling, without further interaction between the matrices. This method is based on priority sampling: the same rows of $A$ and $B$ are selected only when they both contribute significantly to $A^TB$, even though rows are sampled independently within each matrix.

The paper shows that subsampling $O(1/\epsilon^2 \delta)$ rows of $A$ and $B$ suffices to achieve $\epsilon$ error in Frobenius norm with probability $1-\delta$, matching the best known result from dense linear sketches. This paper demonstrates one application of their proposed algorithm in sketched linear regression. Lastly, the paper empirically compares the effectiveness of coordinated random subsampling against linear sketches in terms of approximation accuracy with varying degrees of sparsity in the input matrices, and in two applications: 1) sketched linear regression, 2) approximation to the attention layer in autoregressive language models.

**Strengths:**

1. I found the presentation of this paper to be very clear and engaging. The problem setting, requirements, and notation are all well-defined, and the core idea is explained thoroughly, making it easy for readers to follow. The proof is presented in a clean and structured manner, enhancing readability.

2. The paper provides strong motivation for studying the problem of computing independent sketches and discusses several potential applications, demonstrating their proposed algorithm in a one important application: sketched linear regression.

3. The setting of preserving sparsity of the inputs and computing independent sketches seems to be novel.

**Weaknesses:**

1. One concern I have regarding the experiments is that while vector quantization—a nonlinear compression technique—has been widely studied and applied in practice for approximating the computation of the key matrix in the attention layer, it remains unclear whether using linear compression techniques, such as approximate matrix products, to approximate $QK^T$ or just the key matrix $K$ could degrade model performance significantly in downstream applications. I suggest that the authors cite works that empirically investigate this issue (if there is any) or include explicit discussion and caveats when discussing the use of approximate matrix products for approximating the attention layer. The application discussed in Section 1.4 on multi-vector information retrieval seems more reasonable and well-suited for demonstrating the applicability of the proposed algorithm in experiments.

2. Additionally, some aspects of the experiment section need clarification. For instance, does the x-axis labeled “Sketch Size” in Figures 1-5 represent the percentage of rows subsampled from the input matrices? What does “Layer x” in Figures 4 and 5 refer to? Furthermore, what architecture is used for the autoregressive language model in the experiment? Providing these details would enhance the clarity of the experimental setup.

3. It would also be helpful to include empirical comparisons of memory usage between linear sketches (e.g., JL sketches) and the proposed method.

4. Minor issues: Bad citation in line 109. “is include as” => “is included as” in line 324.

5. To improve clarity for readers, it would be helpful to include a description of $\tau_A$ in the output of Algorithm 1, explaining that it represents the threshold used to determine whether a row in the input matrix $A$ should be retained.

**Questions:**

Why is Theorem 3 presented as a constant probability guarantee rather than a high-probability guarantee? What specific challenges arise in deriving a high-probability guarantee in this context? Additionally, what makes it difficult to achieve a multiplicative $(1+\epsilon)$ guarantee instead of the additive error guarantee stated in the footnote on page 4?

In the context of sketched linear regression problems, the guarantee provided by Theorem 3 is relatively weak. A discussion on the challenges of obtaining a stronger guarantee would be valuable and could provide useful insights for future work.

---

> ### Author Response · Authors · 2024-11-19
> **Response to Reviewer St9U's Comments**
>
> Thanks for the specific suggestions and comments to help improve clarity. We reply to the larger comments/questions below.
>
> ---
>
> ***
>
> 1- While we focus on the abstract problem of matrix product sketching, we acknowledge the importance of understanding how such approximations impact model performance in downstream tasks. To address this, we will cite relevant works, including [1], which uses projection matrices (linear compression) to reduce activation dimensionality, and [2], which applies a Quantized JL Transform {SimHash) to compress the key-value cache in a transformer-based autoregressive language model, where 'key' represents encoded past inputs and 'value' represents their associated outputs.
>
> [1] Sakr et al. ("ESPACE: Dimensionality Reduction of Activations for Model Compression")
>
> [2] Zandieh et al. ("QJL: 1-Bit Quantized JL Transform for KV Cache Quantization with Zero Overhead")
>
> Additionally, we believe that empirically evaluating matrix sketching methods in the context of multi-vector retrieval is an important next step for future work. Compression schemes like product quantization have already been successfully applied in this setting (see, e.g. [3]).
>
> [3] Dhulipala et al. ("MUVERA: Multi-Vector Retrieval via Fixed Dimensional Encodings")
>
> ---
>
> 2- The x-axis labeled "Sketch Size" in Figures 1–5 is the compression ratio of the sketch, specifically the total number of bits requires to store S(A) and S(B) divided by the total number of bits required to store $A$ and $B$. For a JL sketch with $k$ rows of length $d$, the storage complexity is $k \times d \times {64}$ bits, since numbers are stored using doubles. For a sampling based sketch with k rows that have s non-zeros each, the storage complexity is $k\times s \times (64+32)$ bits, since we used 64 bits to store the value of every entry and a $32$ bit int to store its index.
>
> The term "Layer $k$" in Figures 4 and 5 refers to the individual layers (hidden layers) of the Transformer architecture in the LLaMA 2 model. In our experiments, we used a fine-tuned version of `LLaMA 2` (`meta-llama/Llama-2-7b-chat-hf`), which consists of 32 layers. Each layer features multi-head self-attention and feedforward mechanisms. Notably, the $K, V$, and $Q$ matrices in the higher layers tend to have entries with greater differences between values compared to the lower layers, which impacts the behavior of the sketching techniques.
>
> ---
>
> 3- When referring to memory usage, do you mean sketch size? This is the main quantity we compare in the plots currently. In terms of memory usage during the computation of sketches, the difference between the methods is minimal. While one might expect JL to have higher memory requirements due to the multiplication with a dense $k \times n$ matrix, this matrix can be generated dynamically, column by column. As a result, both methods only incur a memory overhead of $O(k)$ when computing the sketch, where $k$ represents the final sketch size.
>
> ---
>
> ***
>
> **Questions**:
>
> 1- We stated Theorem 3 as a constant probability guarantee for simplicity, but the dependence on the failure probability $\delta$ is identical to Theorem 2: we can obtain the result with probability $1-\delta$ if we pay an additional $\frac{1}{\delta}$ multiplicative factor in the sample complexity. In retrospect, this does not add to the theorem’s complexity by much, so we will update it to include this dependence.
>
> This is a good question regarding the feasibility of obtaining a multiplicative $(1 + \epsilon)$ guarantee. Upon further thought, we believe that an additive error dependence is likely necessary for any sampling-based algorithm that sketches $b$ and $A$ independently. We will add a detailed discussion and formal argument to the next version of the paper.
>
> To illustrate this, consider the case where $b$ exactly equals $Ax^*$ for some ground truth $x^*$, so achieving a multiplicative error guarantee would require zero error (since $||Ax^{\*} - b|| = 0$). Now, suppose $A$ is a matrix of height $d + m$, containing every row of the $d \times d$ identity matrix once, but with the first row (the first standard basis vector) appearing $m$ additional times. Finding $x^*$ in this case requires identifying all entries of $b$ that correspond to the $d$ identity rows of $A$. However, if $b$ is sampled independently of $A$, an algorithm has no way of identifying these specific entries, making a multiplicative guarantee unachievable in this setting. This limitation is in strong contrast to scenarios where $b$ is sampled with probabilities that are dependent on $A$.
>
> ---
>
> 2- With the above in mind, it is interesting to ask if there is a method that can both take advantage of sparsity (like our method) but achieve the stronger multiplicative regression guarantee of JL. It seems that doing so will require looking beyond direct “sample and solve” regression methods.

---

> > ### Comment · Reviewer_St9U · 2024-11-26
> > **Follow up on author response**
> >
> > Thank you very much for the answer. While this paper mainly applies existing techniques, the setting it considers is new and well motivated. I will keep my score.

---

### Meta-Review · Area_Chair_tYbq · 2024-12-19

**Metareview:**

This paper explores the approximative computation of the matrix product  $A^\top B$  using sketching techniques, particularly in scenarios where the sketches of $ A $ and $ B $ must be computed independently.
The authors demonstrate that coordinated random sampling-based methods are particularly effective in this setting.
Applications to distributed linear regression and the approximative computation of attention matrices are then considered.

The reviewers reached a consensus that the paper is well-written and that the proposed method, while simple, is efficient.
As a result, I recommend accepting the paper, despite the fact that the theoretical analysis provided is relatively standard.

It would be beneficial if the authors further elaborated on Reviewer NkFE’s comment and reorganized parts of the text to better highlight the technical challenges and distinctions encountered in this work, thereby enhancing the readers' understanding of the contribution of this work.

**Additional Comments On Reviewer Discussion:**

The reviewers raised the following concerns:

- Clarifications on empirical results and settings, etc.: Reviewer St9U requested clarification on the empirical results, while Reviewers Ah4Z and NkFE raise the concern on settings where communication overhead is significant. Additionally, Reviewers St9U and Ah4Z raised questions about the theoretical results, specifically regarding the form of additive (instead of multiplicative) error.
These concerns were successfully addressed by the authors during the rebuttal phase.
- Standard nature of the proposed techniques and theoretical analysis: **All** reviewers noted that the proposed techniques and analysis are relatively standard.
While this concern cannot be easily addressed, the reviewers appear generally satisfied with the authors' response on this matter.

I have carefully considered all of these points in reaching my final decision.

---

### Decision · Program_Chairs · 2025-01-22

Accept (Poster)